# Active Bayesian Causal Inference

**Christian Toth**
TU Graz

**Lars Lorch**
ETH Zürich

**Christian Knoll**
TU Graz

**Andreas Krause**
ETH Zürich

**Franz Pernkopf**
TU Graz

**Robert Peharz**[*]
TU Graz

**Julius von Kügelgen**[*]
MPI for Intelligent Systems, Tübingen
University of Cambridge

## Abstract

Causal discovery and causal reasoning are classically treated as separate and consecutive tasks: one first infers the causal graph, Causal discovery and causal reasoning are classically treated as separate and consecutive tasks: one first infers the causal graph, and then uses it to estimate causal effects of interventions. However, such a two-stage approach is uneconomical, especially in terms of actively collected interventional data, since the causal query of interest may not require a fully-specified causal model. From a Bayesian perspective, it is natural to treat a causal query (e.g., the causal graph or some causal effect) as subject to posterior inference while other unobserved quantities ought to be marginalized out. In this work, we propose Active Bayesian Causal Inference (ABCI), a *fully-Bayesian active learning framework for integrated causal discovery and reasoning*, which jointly infers a posterior over causal models and queries of interest. ABCI sequentially designs experiments that are maximally informative about the target causal query, collects the corresponding interventional data, and updates the Bayesian beliefs to choose the next experiment. Through simulations, we demonstrate that our approach is more data-efficient than several baselines that only focus on learning the full causal graph. This allows us to accurately learn downstream causal queries from fewer samples while providing well-calibrated uncertainty estimates for the quantities of interest.

## 1 Introduction

Causal reasoning, that is, answering causal queries such as the effect of a particular intervention, is a fundamental scientific quest [3, 28, 31, 39]. A rigorous treatment of this quest requires a reference causal model, typically consisting at least of (i) a causal diagram, or directed acyclic graph (DAG), capturing the qualitative causal structure between a system's variables [45] and (ii) a joint distribution that is Markovian w.r.t. this causal graph [62]. Other frameworks additionally model (iii) the functional dependence of each variable on its causal parents in the graph [46, 70]. If the graph is not known from domain expertise, causal discovery aims to infer it from data [38, 62]. However, given only passively-collected observational data and no assumptions on the data-generating process, causal discovery is limited to recovering the Markov equivalence class (MEC) of DAGs implying the conditional independences present in the data [62]. Additional assumptions like linearity can render the graph identifiable [29, 49, 59, 71] but are often hard to falsify, thus leading to risk of misspecification. These shortcomings motivate learning from experimental (interventional) data, which enables recovering the true causal structure [12, 13, 24]. Since obtaining interventional data is costly in practice, we study the active learning setting, in which we sequentially design and perform interventions that are most informative for the target causal query [1, 19, 24, 25, 40, 66].

---

[*]Shared last author.
Correspondence to: {`christian.toth,robert.peharz`}`@tugraz.at`, `jvk@tue.mpg.de`
NeurIPS 2022 main conference paper version available at https://openreview.net/forum?id=r0bjBULkyz.

NeurIPS 2022 Workshop on neuro Causal and Symbolic AI (nCSI 2022).

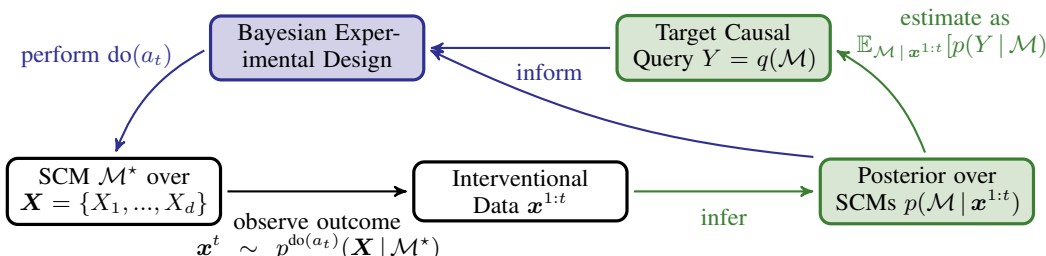

Figure 1: **Overview of the Active Bayesian Causal Inference (ABCI) framework.** At each time step $t$, we use Bayesian experimental design based on our current beliefs to choose a maximally informative intervention $a_t$ to perform. We then collect a finite data sample from the interventional distribution induced by the environment, which we assume to be described by an unknown structural causal model (SCM) $\mathcal{M}^\star$ over a set of observable variables $\boldsymbol{X}$. Given the interventional data $\boldsymbol{x}^{1:t}$ collected from the true SCM $\mathcal{M}^\star$ and a prior distribution over the model class of consideration, we infer the posterior over a target causal query $Y = q(\mathcal{M})$ that can be expressed as a function of the causal model. For example, we may be interested in the graph (causal discovery), the presence of certain edges (partial causal discovery), the full SCM (causal model learning), a collection of interventional distributions or treatment effects (causal reasoning), or any combination thereof.

Classically, causal discovery and reasoning are treated as separate, consecutive tasks that are studied by different communities. Prior work on experimental design has thus focused either purely on causal reasoning—that is, how to best design experimental studies if the causal graph is known?—or purely on causal discovery, whenever the graph is unknown [27, 49]. In the present work, we consider the more general setting in which we are interested in performing causal reasoning but do not have access to a reference causal model a priori. In this case, causal discovery can be seen as a means to an end rather than as the main objective. Focusing on actively learning the *full* causal model to enable subsequent causal reasoning can thus be disadvantageous for two reasons. First, wasting samples on learning the full causal graph is suboptimal if we are only interested in specific aspects of the causal model. Second, causal discovery from small amounts of data entails significant epistemic uncertainty—for example, incurred by low statistical test power or multiple highly-scoring DAGs—which is not taken into account when selecting a single reference causal model [2, 16].

In this work, we propose *Active Bayesian Causal Inference* (ABCI), a fully-Bayesian framework for integrated causal discovery and reasoning with experimental design. The basic approach is to put a Bayesian prior over the causal model class of choice, and to cast the learning problem as Bayesian inference over the model posterior. Given the unobserved causal model, we formalize causal reasoning by introducing the *target causal query*, a function of the causal model that specifies the set of causal quantities we are interested in. The model posterior together with the query function induce a *query posterior*, which represents the result of our Bayesian learning procedure. It can be used, e.g., in downstream decision tasks or to derive a MAP solution or suitable expectation. To learn the query posterior, we follow the Bayesian optimal experimental design approach [8, 33] and sequentially choose admissible interventions on the true causal model that are most informative about our target query w.r.t. our current beliefs. Given the observed data, we then update our beliefs by computing the posterior over causal models and queries and use them to design the next experiment.

## 2 Active Bayesian Causal Inference (ABCI) Framework

In this section, we first introduce the ABCI framework in generality and formalize its main concepts and distributional components, which are illustrated in Fig. 1. In Appx. A, we then describe our particular instantiation of ABCI for the class of causally sufficient nonlinear additive Gaussian noise models.

**Causal Model.** To treat causality in a rigorous way, we first need to postulate a mathematically well-defined causal model. Historically hard questions about causality can then be reduced to epistemic questions, that is, what and how much is known about the causal model. A prominent type of causal model is the structural causal model (SCM) [46]. From a Bayesian perspective, an SCM can be viewed as a hierarchical data-generating process involving latent random variables.

**Definition 1** (SCM). An SCM $\mathcal{M}$ over observed endogenous variables $\boldsymbol{X} = \{X_1, \ldots, X_d\}$ and unobserved exogenous variables $\boldsymbol{U} = \{U_1, \ldots, U_d\}$ consists of structural equations, or mechanisms,

$$X_i := f_i(\mathbf{Pa}_i, U_i), \qquad \text{for} \quad i \in \{1, \ldots, d\}, \tag{2.1}$$

which assign the value of each $X_i$ as a deterministic function $f_i$ of its direct causes, or causal parents, $\mathbf{Pa}_i \subseteq \boldsymbol{X} \setminus \{X_i\}$ and $U_i$; and a joint distribution $p(\mathbf{U})$ over the exogenous variables.

Associated with each SCM is a directed causal graph $G$ with vertices $\boldsymbol{X}$ and edges $X_j \to X_i$ if and only if $X_j \in \mathbf{Pa}_i$, which we assume to be acyclic. Any acyclic SCM then induces a unique observational distribution $p(\boldsymbol{X} \,|\, \mathcal{M})$ over the endogenous variables $\boldsymbol{X}$, which is obtained as the pushforward measure of $p(\boldsymbol{U})$ through the causal mechanisms in Eq. (2.1).

**Interventions.** A crucial aspect of causal models such as SCMs is that they also model the effect of *interventions*—external manipulations to one or more of the causal mechanisms in Eq. (2.1)—which, in general, are denoted using Pearl's do-operator [46] as $\mathrm{do}(\{X_i = \tilde{f}_i(\mathbf{Pa}_i, U_i)\}_{i \in \mathcal{I}})$ with $\mathcal{I} \subseteq [d]$ and suitably chosen $\tilde{f}_i(\cdot)$. An intervention leads to a new SCM, the so-called interventional SCM, in which the relevant structural equations in Eq. (2.1) have been replaced by the new, manipulated ones. The interventional SCM thus induces a new distribution over the observed variables, the so-called interventional distribution, which is denoted by $p^{\mathrm{do}(a)}(\boldsymbol{X} \,|\, \mathcal{M})$ with $a$ denoting the (set of) intervention(s) $\{X_i = \tilde{f}_i(\mathbf{Pa}_i, U_i)\}_{i \in \mathcal{I}}$. Causal effects, that is, expressions like $\mathbb{E}[X_j | \mathrm{do}(X_i = 3)]$, can then be derived from the corresponding interventional distribution via standard probabilistic inference.

**Being Bayesian with Respect to Causal Models.** The main epistemic challenge for causal reasoning stems from the fact that the true causal model $\mathcal{M}^\star$ is not or not completely known. The canonical response to such epistemic challenges is a Bayesian approach: place a prior $p(\mathcal{M})$ over causal models, collect data $\mathcal{D}$ from the true model $\mathcal{M}^\star$, and compute the posterior via Bayes rule:

$$p(\mathcal{M} \,|\, \mathcal{D}) = \frac{p(\mathcal{D} \,|\, \mathcal{M})\, p(\mathcal{M})}{p(\mathcal{D})} = \frac{p(\mathcal{D} \,|\, \mathcal{M})\, p(\mathcal{M})}{\int p(\mathcal{D} \,|\, \mathcal{M})\, p(\mathcal{M})\, \mathrm{d}\mathcal{M}} \,. \tag{2.2}$$

A full Bayesian treatment over $\mathcal{M}$ is computationally delicate, to say the least. We require a way to parameterise the class of models $\mathcal{M}$ while being able to perform posterior inference over this model class. In this paper, we present a fully Bayesian approach for flexibly modelling nonlinear relationships (Appx. A).

**Bayesian Causal Inference.** In the causal inference literature, the tasks of causal discovery and causal reasoning are typically considered separate problems. The former aims to learn (parts of) the causal model $\mathcal{M}^\star$, typically the causal graph $G^\star$, while the latter assumes that the relevant parts of $\mathcal{M}^\star$ are already known and aims to identify and estimate some query of interest, typically using only observational data. This separation suggests a two-stage approach of first performing causal discovery and then fixing the model for subsequent causal reasoning. From the perspective of uncertainty quantification and active learning, however, this distinction is unnatural because intermediate, unobserved quantities like the causal model do not contribute to the epistemic uncertainty in the final quantities of interest. Instead, we define a causal query function $q$, which specifies a *target causal query* $Y = q(\mathcal{M})$ as a function of the causal model $\mathcal{M}$. This view thus subsumes and generalises causal discovery and reasoning into a unified framework. For example, possible causal queries are:

*Causal Discovery:* $\qquad Y = q_{\mathrm{CD}}(\mathcal{M}) = G$, that is, learning the full causal graph $G$;

*Causal Model Learning:* $\quad Y = q_{\mathrm{CML}}(\mathcal{M}) = \mathcal{M}$, that is, learning the full SCM $\mathcal{M}$;

*Causal Reasoning:* $\qquad Y = q_{\mathrm{CR}}(\mathcal{M}) = \{X_j^{\mathrm{do}(\boldsymbol{X}_{\mathcal{I}(j)} = \boldsymbol{\psi}_j)}\}_{j \in \mathcal{J}}$, that is, learning a set of interventional variables $X_j$ induced by $\mathcal{M}$ under $\mathrm{do}(\boldsymbol{X}_{\mathcal{I}(j)} = \boldsymbol{\psi}_j)$.

Given a causal query, Bayesian inference naturally extends to our learning goal, the *query posterior*:

$$p(Y \,|\, \mathcal{D}) = \int p(Y \,|\, \mathcal{M})\, p(\mathcal{M} \,|\, \mathcal{D})\, \mathrm{d}\mathcal{M} = \mathbb{E}_{\mathcal{M} \,|\, \mathcal{D}}[\, p(Y \,|\, \mathcal{M})\,] \,. \tag{2.3}$$

Evidently, computing Eq. (2.3) constitutes a hard computational problem in general, as we need to marginalise out the causal model. In Appx. A, we introduce a practical implementation for a restricted causal model class, informed by this challenge.

**Active Learning with Sequential Interventions.** Rather than collect a large observational dataset, we seek to leverage experimental data, which can help resolve some of the aforementioned identifiability issues and facilitate learning our target causal query more quickly, even if the model is identifiable. Since obtaining experimental data is costly in practice, we study the active learning setting in which we sequentially design experiments in the form of interventions $a_t$. At each time step $t$, the outcome

of this experiment $a_t$ is a batch $\boldsymbol{x}^t$ of $N_t$ i.i.d. observations from the true interventional distribution:

$$\boldsymbol{x}^t = \{\boldsymbol{x}^{t,n}\}_{n=1}^{N_t}, \qquad \boldsymbol{x}^{t,n} \overset{\text{i.i.d.}}{\sim} p^{\text{do}(a_t)}(\boldsymbol{X} \,|\, \mathcal{M}^\star) \tag{2.4}$$

Crucially, we design the experiment $a_t$ to be *maximally informative* about our target causal query $Y$. In our Bayesian setting, this is naturally formulated as maximising the myopic information gain from the next intervention, that is, the mutual information between $Y$ and the outcome $\boldsymbol{X}^t$ [8, 33]:

$$\max_{a_t} \text{I}(Y; \boldsymbol{X}^t \,|\, \boldsymbol{x}^{1:t-1}) \tag{2.5}$$

where $\boldsymbol{X}^t$ follows the predictive interventional distribution of the Bayesian causal model ensemble at time $t - 1$ under intervention $a_t$, which is given by

$$\boldsymbol{X}^t \sim p^{\text{do}(a_t)}(\boldsymbol{X} \,|\, \boldsymbol{x}^{1:t-1}) \propto \int p^{\text{do}(a_t)}(\boldsymbol{X} \,|\, \mathcal{M})\, p(\mathcal{M} \,|\, \boldsymbol{x}^{1:t-1})\, \text{d}\mathcal{M}. \tag{2.6}$$

By maximising Eq. (2.5), we collect experimental data and infer our target causal query $Y$ in a highly efficient, goal-directed manner.

## 3 Experiments

**Setup.** We evaluate ABCI by inferring the query posterior on synthetic ground-truth SCMs using several different experiment selection strategies. Specifically, we design experiments w.r.t. $U_{\text{CD}}$ (causal discovery), $U_{\text{CML}}$ (causal model learning), and $U_{\text{CR}}$ (causal reasoning); see Appx. A.2. We compare against baselines which (i) only sample from the observational distribution (OBS) or (ii) pick an intervention target $j$ uniformly at random from $[d] \cup \{\varnothing\}$ and set $X_j = 0$ (RAND FIXED, a weak random baseline used in prior work) or draw $X_j \sim \mathcal{U}(-7, 7)$ (RAND) if $X_j \neq \varnothing$. All methods follow our Bayesian GP-DiBS-ABCI approach from Appx. A. We sample ground truth SCMs over random scale-free graphs [5] of size $d = 20$, with mechanisms and noise variances drawn from our model prior in Eq. (A.4). In Appx. H, we report additional results for both scale-free and Erdős Renyi random graphs over $d = 10$ resp. $d = 20$ variables. For specific prior choices and simulation details, see Appx. E.

**Metrics.** As ABCI infers a posterior over the target query $Y$, a natural evaluation metric is the Kullback-Leibler divergence (KLD) between the true query distribution and the inferred query posterior, $\text{KL}(p(Y \,|\, \mathcal{M}^\star) \,||\, p(Y \,|\, \boldsymbol{x}^{1:t}))$. We report **Query KLD**, a KLD estimate for target interventional distributions ($q_{\text{CR}}$). As a proxy for the KLD of the SCM posterior ($q_{\text{CML}}$), we report the average KLD across all single node interventional distributions $\{p^{\text{do}(X_i=\psi)}(\boldsymbol{X})\}_{i=1}^d$, with $\psi \sim \mathcal{U}(-7, 7)$ (**Average I-KLD**). We also report the expected structural Hamming distance [11], **ESHD** $= \mathbb{E}_{G \,|\, \boldsymbol{x}^{1:t}}[\text{SHD}(G, G^\star)]$, a commonly used causal discovery metric, and the *area under the precision recall curve* (**AUPRC**). See Appx. G for further details.

**Causal Discovery and SCM Learning (Fig. 2).** In our first experiment, we find that all ABCI-based methods are able to meaningfully learn from small amounts of data, which validates our Bayesian approach. Moreover, performing targeted interventions using experimental design indeed improves performance compared to uninformed experimentation (OBS, RAND FIXED, RAND). Notably, the stronger random baseline (RAND), which also randomises over intervention values, performs well in the considered setting. As expected by the theoretical grounding of the information gain utilities, $U_{\text{CD}}$ identifies the true graph the fastest (as measured by ESHD), whereas $U_{\text{CML}}$ exhibits good scores across all metrics. Further details are given in the caption of Fig. 2.

**Learning Interventional Distributions (Fig. 3).** In our second experiment, we investigate ABCI's causal reasoning capabilities by randomly sampling ground-truth SCMs as described above over the fixed graph shown in Fig. 3 (right), which is *not* known to the methods. Our target query is the set of interventional random variables, or "distributional treatment effects", $X_5^{\text{do}(X_3=\psi)}$ for treatments $\psi \sim \mathcal{U}[2, 5]$. The results show that our informed experiment selection strategies significantly outperform the baselines at causal reasoning as measured by the Query KLD. In accordance with the results from Fig. 2 and considering that, once we know the true SCM, we can compute any causal quantity of interest, $U_{\text{CML}}$ seems to provide a reasonable experimental strategy in case the causal query of interest is *not* known a priori. However, our results indicate that if we *do* know our query of interest, then $U_{\text{CR}}$ provides a more efficient experiment design strategy for its estimation, even when the treatment variable of interest is not directly intervenable. In this case, the task is indeed more difficult, as highlighted by the larger Query KLD values across all considered methods.

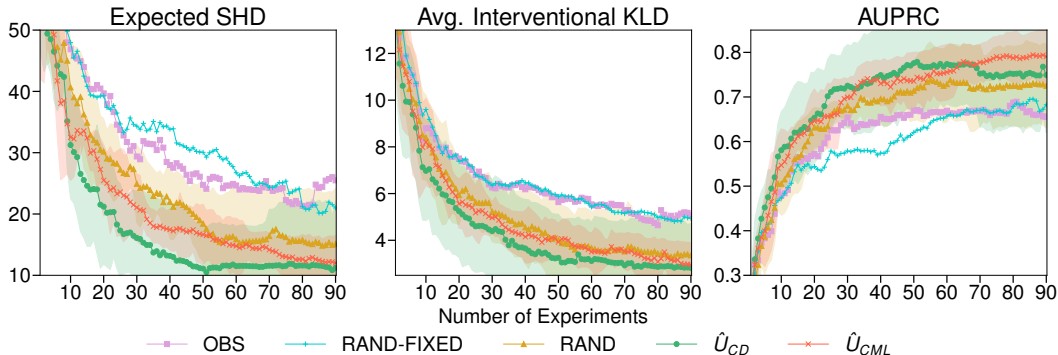

Figure 2: **Causal Discovery and SCM Learning.** Comparison of experimental design strategies for causal discovery ($U_{\text{CD}}$) and causal model learning ($U_{\text{CML}}$) with random and observational baselines on simulated ground truth models with 20 nodes. We initialise all methods with 50 observational samples, and then perform experiments with a batch size of $N_t = 5$. Lines and shaded areas show means and 95% confidence intervals (CIs) across 15 runs (5 randomly sampled ground-truth SCMs with 3 restarts per SCM). CIs for OBS and RAND FIXED baselines are not shown to aid readability; see Fig. 6 in Appx. H for the full figure. **(a) ESHD.** Both our objectives significantly outperform the observational and random baselines. **(b) Average I-KLD.** $U_{\text{CD}}$ significantly outperforms the baselines, whereas $U_{\text{CML}}$ performs only marginally better than RAND. **(c) AUPRC.** Both our strategies perform consistently better than the uninformed selection strategies.

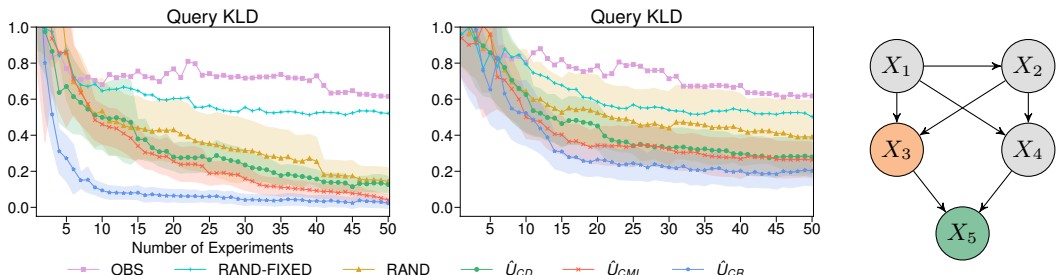

Figure 3: **Learning Interventional Distributions.** (left) Comparison of different methods w.r.t. learning a set of interventional variables $X_5^{\text{do}(X_3=\psi)}$ with $\psi \sim \mathcal{U}[2,5]$ on simulated ground truth models with fixed causal graph (right). We initialise all methods with 5 observational samples, and then perform experiments with a batch size of $N_t = 3$. Lines and shaded areas show means and 95% confidence intervals (CIs) across 30 runs (10 randomly sampled ground truth SCMs with 3 restarts each). CIs for OBS and RAND FIXED baselines are not shown to aid readability; see Figs. 9 and 10 in Appx. H for the full figure. **(a) All nodes actionable.** $U_{\text{CR}}$ significantly outperforms all other methods as expected. $U_{\text{CML}}$ performs second best which, in conjunction with the results from Fig. 2, suggests that $U_{\text{CML}}$ yields a solid base model for performing downstream causal inference tasks. **(b) $X_3$ not actionable.** In this setting, where we cannot directly intervene on the treatment variable of interest, $U_{\text{CR}}$ clearly outperforms all other methods for $\geq 10$ experiments.

## 4 Conclusion

We have introduced ABCI, an Active Bayesian Inference framework for causal queries. The main conceptual advantages of the ABCI framework are that it is *flexible* and *principled*, providing a fresh perspective on the classical divide between causal discovery and reasoning: sometimes, the main objective may be to foster scientific understanding by uncovering the causal structure, while other times, causal discovery may only be a means to an end to support causal reasoning.

In our experiments, we have made several assumptions to facilitate tractable inference and showcase the ABCI framework on a relatively simple data-generating process. Relaxing these assumptions are promising directions for future works, in particular to include heteroscedastic noise, unobserved confounding, and cyclic relationships. Moreover, replacing Gaussian process models with deep generative models is a natural way improve the expressivity of the model, potentially leading to "fully-fledged" neuro-causal models.

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

# Appendices

## Table of Contents

# A  Tractable ABCI for Nonlinear Additive Noise Models

Having described the general ABCI framework and its conceptual components, we now detail how to instantiate ABCI for a flexible model class that still allows for tractable, approximate inference. This requires us to specify (i) the class of causal models we consider in Eq. (2.1), (ii) the types of interventions $a_t$ we consider at each step and the corresponding interventional likelihood in Eq. (2.4), (iii) our prior distribution $p(\mathcal{M})$ over models, (iv) how to perform tractable inference of the model posterior in Eq. (2.2), and finally (v) how to maximise the information gain in Eq. (2.5) for experimental design.

**Model Class and Parametrisation.** In the following, we consider nonlinear additive Gaussian noise models [29] of the form

$$X_i := f_i(\mathbf{Pa}_i) + U_i, \qquad \text{with} \qquad U_i \sim \mathcal{N}(0, \sigma_i^2) \qquad \text{for} \quad i \in \{1, \ldots, d\}, \qquad \text{(A.1)}$$

where the $f_i$'s are smooth, nonlinear functions and the $U_i$'s are assumed to be mutually independent. The latter corresponds to the assumption of causal sufficiency, or no hidden confounding. Any model $\mathcal{M}$ in this model class can be parametrised as a triple $\mathcal{M} = (G, \boldsymbol{f}, \boldsymbol{\sigma}^2)$, where $G$ is a causal DAG, $\boldsymbol{f} = (f_1, \ldots, f_d)$ is a vector of functions defined over the parent sets implied by $G$, and $\boldsymbol{\sigma}^2 = (\sigma_1^2, \ldots, \sigma_d^2)$ contains the Gaussian noise variances. Provided that the $f_i$ are nonlinear and not constant in any of their arguments, the model is identifiable almost surely [29, 50].

**Interventional Likelihood.** We support the realistic setting where only a subset $\boldsymbol{W} \subseteq \boldsymbol{X}$ of all variables are actionable, that is, can be intervened upon.[2] We consider hard interventions of the form $\mathrm{do}(a_t) = \mathrm{do}(\boldsymbol{X}_\mathcal{I} = \boldsymbol{x}_\mathcal{I})$ that fix a subset $\boldsymbol{X}_\mathcal{I} \subseteq \boldsymbol{W}$ to a constant $\boldsymbol{x}_\mathcal{I}$. Due to causal sufficiency, the interventional likelihood under such hard interventions $a_t$ factorises over the causal graph $G$ and is given by the g-formula [52] or truncated factorisation [62]:

$$p^{\mathrm{do}(a_t)}(\boldsymbol{X} \,|\, G, \boldsymbol{f}, \boldsymbol{\sigma}^2) = \mathbb{I}\{\boldsymbol{X}_\mathcal{I} = \boldsymbol{x}_\mathcal{I}\} \prod_{j \notin \mathcal{I}} p(X_j \,|\, f_j(\mathbf{Pa}_j^G), \sigma_j^2). \qquad \text{(A.2)}$$

The last term in Eq. (A.2) is given by $\mathcal{N}(X_j \,|\, f_j(\mathbf{Pa}_j^G), \sigma_j^2)$, due to the Gaussian noise assumption. Let $\boldsymbol{x}^{1:t}$ be the entire dataset, collected up to time $t$. The likelihood of $\boldsymbol{x}^{1:t}$ is then given by

$$p(\boldsymbol{x}^{1:t} \,|\, G, \boldsymbol{f}, \boldsymbol{\sigma}^2) = \prod_{\tau=1}^{t} p^{\mathrm{do}(a_\tau)}(\boldsymbol{x}^\tau \,|\, G, \boldsymbol{f}, \boldsymbol{\sigma}^2) = \prod_{\tau=1}^{t} \prod_{n=1}^{N_t} p^{\mathrm{do}(a_\tau)}(\boldsymbol{x}^{\tau,n} \,|\, G, \boldsymbol{f}, \boldsymbol{\sigma}^2). \qquad \text{(A.3)}$$

**Structured Model Prior.** To specify our prior, we distinguish between root nodes $X_i$, for which $\mathbf{Pa}_i = \varnothing$ and thus $f_i = \text{const}$, and non-root nodes $X_j$. For a given causal graph $G$, we denote the index set of root nodes by $\mathbf{R}(G) = \{i \in [d] : \mathbf{Pa}_i^G = \varnothing\}$ and that of non-root nodes by $\mathbf{NR}(G) = [d] \setminus \mathbf{R}(G)$. We then place the following structured prior over SCMs $\mathcal{M} = (G, \boldsymbol{f}, \boldsymbol{\sigma}^2)$:

$$p(\mathcal{M}) = p(G)\, p(\boldsymbol{f}, \boldsymbol{\sigma}^2 \,|\, G) = p(G) \prod_{i \in \mathbf{R}(G)} p(f_i, \sigma_i^2 \,|\, G) \prod_{j \in \mathbf{NR}(G)} p(f_j \,|\, G) p(\sigma_j^2 \,|\, G). \qquad \text{(A.4)}$$

Here, $p(G)$ is a prior over graphs and $p(\boldsymbol{f}, \boldsymbol{\sigma}^2 \,|\, G)$ is a prior over the functions and noise variances. We factorise our prior conditional on $G$ as in Eq. (A.4) not only to allow for a separate treatment of root vs. non-root nodes, but also to share priors across similar graphs. Whenever $\mathbf{Pa}_i^{G_1} = \mathbf{Pa}_i^{G_2}$, we set $p(f_i, \sigma_i^2 \,|\, G_1) = p(f_i, \sigma_i^2 \,|\, G_2)$. As a consequence, the posteriors are also shared, which substantially reduces the computational cost in practice (see Appx. F.2 for details). Our prior also encodes the beliefs that $\{f_i, \sigma_i^2\} \perp\!\!\!\perp \{f_{i'}, \sigma_{i'}^2\} \,|\, G$ for $i \neq i' \in [d]$ and that $f_j \perp\!\!\!\perp \sigma_j^2 \,|\, G$ for $j \in \mathbf{NR}(G)$ which is motivated by the principle of independent causal mechanisms [49] and the causal sufficiency assumption. Our specific choices for the different factors on the RHS of Eq. (A.4) are guided by ensuring tractable inference and described in more detail below.

**Model Posterior.** Given collected data $\boldsymbol{x}^{1:t}$, we can update our beliefs and quantify our uncertainty in $\mathcal{M}^\star$ by inferring the posterior $p(\mathcal{M} \,|\, \boldsymbol{x}^{1:t})$ over SCMs $\mathcal{M} = (G, \boldsymbol{f}, \boldsymbol{\sigma}^2)$, which can be written as[3]

---

[2]In principle, the set of actionable variables might even change over time, in which case they are denoted $\boldsymbol{W}_t$.

[3]To avoid further complicating the notation, we write all posteriors and likelihoods in terms of the full data $\boldsymbol{x}^{1:t}$. However, only observations of $X_i$ and $X_j \,|\, \mathbf{Pa}_j^G$ matter for $i \in \mathbf{R}(G)$ and $j \in \mathbf{NR}(G)$.

$$p(\mathcal{M} \mid \boldsymbol{x}^{1:t}) = p(G \mid \boldsymbol{x}^{1:t}) \prod_{i \in \mathbf{R}(G)} p(f_i, \sigma_i^2 \mid \boldsymbol{x}^{1:t}, G) \prod_{j \in \mathbf{NR}(G)} p(f_j, \sigma_j^2 \mid \boldsymbol{x}^{1:t}, G). \quad \text{(A.5)}$$

For root nodes $i \in \mathbf{R}(G)$, posterior inference given the graph is straightforward. We have $f_i = \text{const}$, so $f_i$ can be viewed as the mean of $U_i$. We thus place conjugate normal-inverse-gamma N-$\Gamma^{-1}(\mu_i, \lambda_i, \alpha_i^{\mathrm{R}}, \beta_i^{\mathrm{R}})$ priors on $p(f_i, \sigma_i^2 \mid G)$, which allows us to analytically compute the root node posteriors $p(f_i, \sigma_i^2 \mid \boldsymbol{x}^{1:t}, G)$ in Eq. (A.5) given the hyperparameters $(\boldsymbol{\mu}, \boldsymbol{\lambda}, \boldsymbol{\alpha}^{\mathrm{R}}, \boldsymbol{\beta}^{\mathrm{R}})$ [41].

The posteriors over graphs and non-root nodes $j \in \mathbf{NR}(G)$ are given by

$$p(G \mid \boldsymbol{x}^{1:t}) = \frac{p(\boldsymbol{x}^{1:t} \mid G) \, p(G)}{p(\boldsymbol{x}^{1:t})}, \qquad p(f_j, \sigma_j^2 \mid \boldsymbol{x}^{1:t}, G) = \frac{p(\boldsymbol{x}^{1:t} \mid G, f_j, \sigma_j^2) \, p(f_j, \sigma_j^2 \mid G)}{p(\boldsymbol{x}^{1:t} \mid G)}. \quad \text{(A.6)}$$

Computing these posteriors is more involved and discussed in the following.

## A.1 Addressing Challenges for Posterior Inference with GPs and DiBS

The posterior distributions in Eq. (A.6) are intractable to compute in general due to the marginal likelihood and evidence terms $p(\boldsymbol{x}^{1:t} \mid G)$ and $p(\boldsymbol{x}^{1:t})$, respectively. In the following, we will address these challenges by means of appropriate prior choices and approximations.

**Challenge 1: Marginalising out the Functions.** The marginal likelihood $p(\boldsymbol{x}^{1:t} \mid G)$ reads

$$p(\boldsymbol{x}^{1:t} \mid G) = \int p(\boldsymbol{x}^{1:t} \mid G, f_j, \sigma_j^2) \, p(f_j \mid G) \, p(\sigma_j^2 \mid G) \, \mathrm{d}f_j \, \mathrm{d}\sigma_j^2 \quad \text{(A.7)}$$

and requires evaluating integrals over the function domain. We use Gaussian processes (GPs) [69] as an elegant way to solve this problem, as GPs flexibly model *nonlinear* functions while offering convenient analytical properties. Specifically, we place a $\mathcal{GP}(0, k_j^G(\cdot, \cdot))$ prior on $p(f_j \mid G)$, where $k_j^G(\cdot, \cdot)$ is a covariance function over the parents of $X_j$ with kernel parameters $\boldsymbol{\kappa}_j$. As is common, we refer to $(\boldsymbol{\kappa}_j, \sigma_j^2)$ as the GP-hyperparameters. In addition, we place $\text{Gamma}(\alpha_j^\sigma, \beta_j^\sigma)$ and $\text{Gamma}(\boldsymbol{\alpha}_j^\kappa, \boldsymbol{\beta}_j^\kappa)$ priors on $p(\sigma_i^2 \mid G)$ and $p(\boldsymbol{\kappa}_i \mid G)$ and collect their parameters in $(\boldsymbol{\alpha}^{\mathrm{GP}}, \boldsymbol{\beta}^{\mathrm{GP}})$.

The graphical model underlying all variables and hyperparameters is shown in Fig. 4. For our model class, GPs provide closed-form expressions for the GP-marginal likelihood $p(\boldsymbol{x}^{1:t} \mid G, \sigma_j^2, \boldsymbol{\kappa}_j)$, as well as for the GP posteriors $p(f_j \mid \boldsymbol{x}^{1:t}, G, \sigma_j^2, \boldsymbol{\kappa}_j)$ and the predictive posteriors over observations $p(\boldsymbol{X} \mid \boldsymbol{x}^{1:t}, G, \boldsymbol{\sigma}^2, \boldsymbol{\kappa})$ [69], see Appx. C for details.

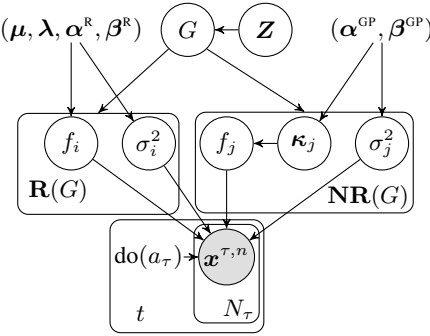

Figure 4: Graphical model of GP-DiBS-ABCI.

**Challenge 2: Marginalising out the GP-Hyperparameters.** While GPs allow for exact posterior inference conditional on a fixed instance of $(\sigma_j^2, \boldsymbol{\kappa}_j)$, evaluating expressions such as $p(f_j \mid \boldsymbol{x}^{1:t}, G)$ requires marginalising out these GP-hyperparameters from the GP-posterior. In general, this is intractable to do exactly, as there is no analytical expression for $p(\sigma_j^2, \boldsymbol{\kappa}_j \mid \boldsymbol{x}^{1:t}, G)$. To tackle this, we approximate such terms using a maximum a posteriori (MAP) point estimate $(\hat{\sigma}_j^2, \hat{\boldsymbol{\kappa}}_j)$ obtained by performing gradient ascent on the unnormalised log posterior

$$\nabla \log p(\sigma_j^2, \boldsymbol{\kappa}_j \mid \boldsymbol{x}^{1:t}, G) = \nabla \log p(\boldsymbol{x}^{1:t} \mid G, \sigma_j^2, \boldsymbol{\kappa}_j) + \nabla \log p(\sigma_j^2, \boldsymbol{\kappa}_j \mid G) \quad \text{(A.8)}$$

according to a predefined update schedule, see Alg. 1. More specifically,

$$p(f_j \mid \boldsymbol{x}^{1:t}, G) = \int p(f_j \mid \boldsymbol{x}^{1:t}, G, \sigma_j^2, \boldsymbol{\kappa}_j) p(\sigma_j^2, \boldsymbol{\kappa}_j \mid \boldsymbol{x}^{1:t}, G) \, \mathrm{d}\sigma_j^2 \, \mathrm{d}\boldsymbol{\kappa}_j \approx p(f_j \mid \boldsymbol{x}^{1:t}, G, \hat{\sigma}_j^2, \hat{\boldsymbol{\kappa}}_j)$$

**Challenge 3: Marginalising out the Causal Graph.** The evidence $p(\boldsymbol{x}^{1:t})$ is given by

$$p(\boldsymbol{x}^{1:t}) = \sum_G p(\boldsymbol{x}^{1:t} \mid G) \, p(G) \quad \text{(A.9)}$$

**Algorithm 1:** GP-DiBS-ABCI for nonlinear additive Gaussian noise models

---

**Input:** # of experiments $T$, batch sizes $\{N_t\}_{t=1}^T$, # of latent particles $M$, # of MC samples $K$, particle resampling schedule $\{r_t\}_{t=1}^T$, hyperparameter update schedule $\{s_t\}_{t=1}^T$
**Output:** Posterior over target causal query $p(Y \,|\, \boldsymbol{x}^{1:T})$

---

$\boldsymbol{z}^0 \sim p(\boldsymbol{Z})$      ▷`sample initial particles; Eq.` (E.12)
**for** $t = 1$ **to** $T$ **do**
    $a_t \leftarrow \arg\max_{a=(\mathcal{I}, \boldsymbol{x}_{\mathcal{I}})} U(a, \boldsymbol{x}^{1:t-1})$      ▷`design experiment; Eq.` (A.11)
    $\boldsymbol{x}^t \leftarrow \{\boldsymbol{x}^{(t,n)} \sim p^{\mathrm{do}(a_t)}(\boldsymbol{X} \,|\, \mathcal{M}^\star)\}_{n=1}^{N_t}$      ▷`perform experiment; Eq.` (2.4)
    $\boldsymbol{z}^t \leftarrow \boldsymbol{z}^{t-1}$
    **if** $r_t$ **then**
       $\boldsymbol{z}^t \leftarrow$ `resample_particles`$(\boldsymbol{z}^t)$      ▷`see Appx.`F
    **end**
    **repeat**
       $\boldsymbol{G} \leftarrow \{G^{(k,m)} \sim p(G \,|\, \boldsymbol{z}_m^t)\}_{k=1}^K {}_{m=1}^M$      ▷`sample graphs; Eq.` (E.11)
       $\boldsymbol{\kappa}, \boldsymbol{\sigma}^2 \leftarrow$ `estimate_hyperparameters`$(\boldsymbol{x}^{1:s_t}, \boldsymbol{G})$      ▷`see Eq.` (A.8)
       $\boldsymbol{z}^t \leftarrow$ `svgd_step`$(\boldsymbol{z}^t, \boldsymbol{x}^{1:t}, \boldsymbol{G}, \boldsymbol{\kappa}, \boldsymbol{\sigma}^2)$      ▷`update latent particles`
    **until** `svgd_convergence`      ▷$\boldsymbol{z}^t$ `now approximate` $p(\boldsymbol{Z} \,|\, \boldsymbol{x}^{1:t})$
**end**

---

and involves a summation over all possible DAGs $G$. This becomes intractable for $d \geq 5$ variables as the number of DAGs grows super-exponentially in the number of variables [53]. To address this challenge, we employ the recently proposed DiBS framework [35]. By introducing a continuous prior $p(\boldsymbol{Z})$ that models $G$ via $p(G \,|\, \boldsymbol{Z})$ and simultaneously enforces acyclicity of $G$, Lorch et al. [35] show that we can efficiently infer the discrete posterior $p(G \,|\, \boldsymbol{x}^{1:t})$ via $p(\boldsymbol{Z} \,|\, \boldsymbol{x}^{1:t})$ as

$$\mathbb{E}_{G \,|\, \boldsymbol{x}^{1:t}} [\phi(G)] = \mathbb{E}_{\boldsymbol{Z} \,|\, \boldsymbol{x}^{1:t}} \left[ \frac{\mathbb{E}_{G \,|\, \boldsymbol{Z}} [\, p(\boldsymbol{x}^{1:t} \,|\, G)\, \phi(G)]}{\mathbb{E}_{G \,|\, \boldsymbol{Z}} [\, p(\boldsymbol{x}^{1:t} \,|\, G)]} \right] \tag{A.10}$$

where $\phi$ is some function of the graph. Since $p(\boldsymbol{Z} \,|\, \boldsymbol{x}^{1:t})$ is a continuous density with tractable gradient estimators, we can leverage efficient variational inference methods such as Stein Variational Gradient Descent (SVGD) for approximate inference [34]. Additional details on DiBS are given in Appx. E.

## A.2 Approximate Bayesian Experimental Design with Bayesian Optimisation

Following § 2, our goal is to perform experiments $a_t$ that are maximally informative about our target query $Y = q(\mathcal{M})$ by maximising the information gain from Eq. (2.5) given our hitherto collected data $\mathcal{D} := \boldsymbol{x}^{1:t-1}$. In Appx. D, we show that this is equivalent to maximising the following utility function:

$$U(a) = H(\boldsymbol{X}^t \,|\, \mathcal{D}) + \mathbb{E}_{\mathcal{M} \,|\, \mathcal{D}} \left[ \mathbb{E}_{\boldsymbol{X}^t, Y \,|\, \mathcal{M}} \left[ \log \mathbb{E}_{\mathcal{M}' \,|\, \mathcal{D}} \left[ p(\boldsymbol{X}^t, Y \,|\, \mathcal{M}') \right] \right] \right], \tag{A.11}$$

where

$$H(\boldsymbol{X}^t \,|\, \mathcal{D}) = \mathbb{E}_{\mathcal{M} \,|\, \mathcal{D}} \left[ \mathbb{E}_{\boldsymbol{X}^t \,|\, \mathcal{M}} \left[ \log \mathbb{E}_{\mathcal{M}' \,|\, \mathcal{D}} \left[ p(\boldsymbol{X}^t \,|\, \mathcal{M}') \right] \right] \right]$$

denotes the differential entropy of the experiment outcome which depends on $a$ and is distributed as in Eq. (2.6). This surrogate objective can be estimated using a nested Monte Carlo estimator as long as we can sample from and compute $p(Y \,|\, \mathcal{M})$, or alternatively, $p(Y \,|\, \boldsymbol{X}^t, G, \mathcal{D})$. Refer to Appx. D for further details. For example, for $q_{\mathrm{CR}}(\mathcal{M}) = X_j^{\mathrm{do}(X_i = \psi)}$ with $\psi \sim p(\psi)$ a distribution over intervention values, we obtain:

$$U_{\mathrm{CR}}(a) = \mathbb{E}_{G \,|\, \mathcal{D}} \big[ \mathbb{E}_{\boldsymbol{X}^t \,|\, G, \mathcal{D}} \big[ - \log \mathbb{E}_{G' \,|\, \mathcal{D}} \left[ p(\boldsymbol{X}^t \,|\, \mathcal{D}, G') \right] \tag{A.12}$$
$$+ \mathbb{E}_\psi \big[ \mathbb{E}_{X_j \,|\, \boldsymbol{X}^t, G, \mathcal{D}}^{\mathrm{do}(X_i = \psi)} \big[ \log \mathbb{E}_{G' \,|\, \mathcal{D}} \big[ p(\boldsymbol{X}^t \,|\, \mathcal{D}, G)\, p^{\mathrm{do}(X_i = \psi)}(X_j \,|\, \boldsymbol{X}^t, G, \mathcal{D}) \big] \big] \big] \big] \big].$$

Importantly, for specific instances of the query function $q(\cdot)$ discussed in § 2, we can derive simpler utility functions than Eq. (A.11). For example, for $q_{\mathrm{CD}}(\mathcal{M}) = G$ and $q_{\mathrm{CML}}(\mathcal{M}) = \mathcal{M}$, we arrive at

$$U_{\mathrm{CD}}(a) = \mathbb{E}_{G \,|\, \mathcal{D}} \left[ \mathbb{E}_{\boldsymbol{X}^t \,|\, G, \mathcal{D}} \left[ \log p(\boldsymbol{X}^t \,|\, \mathcal{D}, G) - \log \mathbb{E}_{G' \,|\, \mathcal{D}} \left[ p(\boldsymbol{X}^t \,|\, \mathcal{D}, G') \right] \right] \right], \tag{A.13}$$

$$U_{\mathrm{CML}}(a) = \mathbb{E}_{\mathcal{M} \,|\, \mathcal{D}} \left[ \mathbb{E}_{\boldsymbol{X}^t \,|\, \mathcal{M}} \left[ \log p(\boldsymbol{X}^t \,|\, \mathcal{M}) - \log \mathbb{E}_{G' \,|\, \mathcal{D}} \left[ p(\boldsymbol{X}^t \,|\, \mathcal{D}, G') \right] \right] \right], \tag{A.14}$$

where the entropy $\mathbb{E}_{\boldsymbol{X}^t \mid \mathcal{M}}[\log p(\boldsymbol{X}^t \mid \mathcal{M})]$ can again be efficiently computed given our modelling choices. For brevity, we defer derivations and estimation details to Appxs. D and E.

Finding the optimal experiment $a_t^* = (\mathcal{I}^*, \boldsymbol{x}_{\mathcal{I}}^*)$ requires jointly optimising the utility function corresponding to our query with respect to (i) the set of intervention *targets* $\mathcal{I}$ and (ii) the corresponding intervention *values* $\boldsymbol{x}_{\mathcal{I}}$. This lends itself naturally to a nested, bi-level optimisation scheme [67]:

$$\mathcal{I}^* \in \arg\max_{\mathcal{I}} U(\mathcal{I}, \boldsymbol{x}_{\mathcal{I}}^*), \quad \text{where} \quad \forall \mathcal{I}: \qquad \boldsymbol{x}_{\mathcal{I}}^* \in \arg\max_{\boldsymbol{x}_{\mathcal{I}}} U(\mathcal{I}, \boldsymbol{x}_{\mathcal{I}}), \tag{A.15}$$

In the above, we first estimate the optimal intervention values for all candidate intervention targets $\mathcal{I}$ and then select the intervention target that yields the highest utility. The intervention target $\mathcal{I}$ may contain multiple variables, which would yield a combinatorial problem. For simplicity, we consider only single-node interventions, $|\mathcal{I}| = 1$. To find $\boldsymbol{x}_{\mathcal{I}}^*$, we employ Bayesian optimisation [36, 37, 60] to efficiently estimate the most informative intervention value $\boldsymbol{x}_{\mathcal{I}}^*$, see Appx. E.

# B   Further Discussion of Related Work

Causal discovery and reasoning have been widely studied in machine learning and statistics [20, 27, 49, 68]. Given an already collected set of observations, there is a large body of literature on learning causal structure, both in the form of a point estimate [7, 23, 48, 59, 62] and a Bayesian posterior [2, 10, 16, 26, 35]. Given a known causal graph, previous work studies how to estimate treatment effects or counterfactuals [46, 55, 57]. When interventional data is yet to be collected, existing work primarily focuses on the specific task of structure learning—without its downstream use. The concept of (Bayesian) active causal discovery was first considered in discrete [40, 66] or linear [9, 43] models with closed-form marginal likelihoods and later extended to nonlinear causal mechanisms [65, 67], multi-target interventions [64], and general models by using hypothesis testing [17] or heuristics [56]. Graph theoretic works give insights on the interventions required for full identifiability [12, 13, 24, 30, 58].

Beyond learning the complete causal graph, few prior works have studied active causal inference. Concurrent work of Tigas et al. [65] considers experimental design for learning a full SCM parameterised by neural networks. There are significant differences to our approach. In particular, our framework (§ 2) is not limited to the information gain over the full model and provides a fully Bayesian treatment of the functions and their epistemic uncertainty (Appx. A). Agrawal et al. [1] consider actively learning a function of the causal graph under budget constraints, though not of the causal mechanisms and only for linear Gaussian models. Conversely, Rubenstein et al. [54] perform experimental design for learning the causal mechanisms after the causal graph has been inferred. Thus, while prior work considers causal discovery and reasoning as separate tasks, ABCI forms an integrated Bayesian approach for learning causal queries through interventions, reducing to previously studied settings in special cases. We further discuss related work in Appx. B.

## C   Background on Gaussian processes

We use Gaussian Processes (GPs) to model mechanisms of non-root nodes $X_i$, i.e., we place a GP prior on $p(f_i \,|\, G)$. In the following, we give some background on GPs and how to compute probabilistic quantities thereof relevant to this work. For further information on GPs we refer the reader to Williams and Rasmussen [69].

A $\mathcal{GP}(m_i(\cdot), k_i^G(\cdot, \cdot))$ is a collection of random variables, any finite number of which have a joint Gaussian distribution, and is fully determined by its mean function $m_i(\cdot)$ and covariance function (or kernel) $k_i^G(\cdot, \cdot)$, where

$$m(\mathbf{x}) = \mathbb{E}[f(\mathbf{x})], \quad \text{and} \quad k(\mathbf{x}, \mathbf{x}') = \mathbb{E}[(f(\mathbf{x}) - m(\mathbf{x}))(f(\mathbf{x}') - m(\mathbf{x}'))]. \tag{C.1}$$

In our experiments, we choose the mean function $m_i(x) \equiv 0$ to be zero and a rational quadratic kernel

$$k_{RQ}(\mathbf{x}, \mathbf{x}') = \kappa_i^o \cdot \left(1 + \frac{1}{2\alpha}(\mathbf{x} - \mathbf{x}')^\top \kappa_i^l (\mathbf{x} - \mathbf{x}')\right)^{-\alpha} \tag{C.2}$$

as our covariance function. Here, $\alpha$ denotes a weighting parameter, $\kappa_i^o$ denotes an output scale parameter and $\kappa_i^l$ denotes a length scale parameter. For the weighting parameter, we use a default value of $\alpha = \log 2 \approx 0.693$. For $\kappa_i^l$ and $\kappa_i^o$ we choose priors according to Appx. E.4. In Section A.1 we summarise both parameters as $\boldsymbol{\kappa}_i = (\kappa_i^o, \kappa_i^l)$.

In this work, we consider Gaussian additive noise models (see Eq. (A.1)). Hence, for a given non-root node $X_i$ in some graph $G$, we have

$$p(X_i \,|\, \mathbf{pa}_i^G, f_i, \sigma_i^2, G) = \mathcal{N}(X_i \,|\, f_i(\mathbf{pa}_i^G), \sigma_i^2) \tag{C.3}$$

where $\mathbf{pa}_i^G$ denotes the parents of $X_i$ in $G$. For some batch of collected data $\boldsymbol{x} = \{x^n\}_{n=1}^N$, let $\boldsymbol{x}_i = (x_i^1, \dots x_i^N)^T$, $\mathbf{pa}_i^G = (\mathbf{pa}_i^{G,1}, \dots, \mathbf{pa}_i^{G,N})$, and $\boldsymbol{K}$ the Gram matrix with entries $K_{m,n} = k_{RQ}(\mathbf{pa}_i^{G,m}, \mathbf{pa}_i^{G,n})$. Then, we can compute the prior marginal log-likelihood, which is needed to compute $p(x^{1:t} \,|\, G)$, in closed form as

$$\log p(\boldsymbol{x}_i \,|\, \mathbf{pa}_i^G, \sigma_i^2, G) = \log \mathbb{E}_{f_i \,|\, G}\left[p(\boldsymbol{x}_i \,|\, \mathbf{pa}_i^G, f_i, \sigma_i^2, G)\right] \tag{C.4}$$

$$= -\frac{1}{2}\boldsymbol{x}_i^T (K + \sigma^2 I)^{-1} \boldsymbol{x}_i - \frac{1}{2}\log |K + \sigma^2 I| - \frac{N}{2}\log 2\pi. \tag{C.5}$$

To predict the function values $f_i(\widetilde{\mathbf{pa}}_i^G)$ at unseen test locations $\widetilde{\mathbf{pa}}_i^G = (\widetilde{\mathbf{pa}}_i^{G,1}, \dots, \widetilde{\mathbf{pa}}_i^{G,\tilde{N}})$ given previously observed data $\boldsymbol{x}$, let $\boldsymbol{K}^\dagger$ be the $(\tilde{N} \times N)$ covariance matrix with entries $K_{m,n}^\dagger = k_{RQ}(\widetilde{\mathbf{pa}}_i^{G,m}, \mathbf{pa}_i^{G,n})$ and $\tilde{\boldsymbol{K}}$ be the $(\tilde{N} \times \tilde{N})$ covariance matrix with entries $\tilde{K}_{m,n} = k_{RQ}(\widetilde{\mathbf{pa}}_i^{G,m}, \widetilde{\mathbf{pa}}_i^{G,n})$. Then, the predictive posterior is multivariate Gaussian

$$p(f_i(\widetilde{\mathbf{pa}}_i^G) \,|\, \widetilde{\mathbf{pa}}_i^G, \boldsymbol{x}, \sigma_i^2, G) = \mathcal{N}(\boldsymbol{\mu}_f, \boldsymbol{\Sigma}_f) \tag{C.6}$$

with mean

$$\boldsymbol{\mu}_f = \boldsymbol{K}^\dagger \left[\boldsymbol{K} + \sigma_i^2 I\right]^{-1} \boldsymbol{x}_i \tag{C.7}$$

and covariance

$$\boldsymbol{\Sigma}_f = \tilde{\boldsymbol{K}} - \boldsymbol{K}^\dagger \left[\boldsymbol{K} + \sigma_i^2 I\right]^{-1} \boldsymbol{K}^\dagger. \tag{C.8}$$

Finally, the marginal posterior over observations $\tilde{\boldsymbol{X}}_i$, which is needed to sample and evaluate candidate experiments in the experimental design process, is given by

$$p(\tilde{\boldsymbol{X}}_i \,|\, \widetilde{\mathbf{pa}}_i^G, \boldsymbol{x}, \sigma_i^2, G) = \mathcal{N}(\boldsymbol{\mu}_{X_i}, \boldsymbol{\Sigma}_{X_i}) \tag{C.9}$$

with mean

$$\boldsymbol{\mu}_{X_i} = \boldsymbol{\mu}_f \tag{C.10}$$

and covariance

$$\boldsymbol{\Sigma}_{X_i} = \boldsymbol{\Sigma}_f + \sigma_i^2 I. \tag{C.11}$$

# D Derivation of the Information Gain Utility Functions

In the following, we provide the derivations for the expressions presented in Section A.2.

## D.1 Information Gain for General Queries

We show that

$$\underset{a_t}{\arg\max}\ \mathrm{I}(Y; \boldsymbol{X}^t \mid \boldsymbol{x}^{1:t-1}) \quad = \quad \underset{a_t}{\arg\max}\ U(a_t) \tag{D.1}$$

for $U(a_t)$ given in Eq. (A.11).

**Proof.** We write the mutual information in the following form

$$\mathrm{I}(Y; \boldsymbol{X}^t \mid \boldsymbol{x}^{1:t-1}) = H(Y \mid \boldsymbol{x}^{1:t-1}) + H(\boldsymbol{X}^t \mid \boldsymbol{x}^{1:t-1}) - H(Y, \boldsymbol{X}^t \mid \boldsymbol{x}^{1:t-1}). \tag{D.2}$$

In the above, we expand the joint entropy of experiment outcome and query as

$$H(Y, \boldsymbol{X}^t \mid \boldsymbol{x}^{1:t-1}) = -\mathbb{E}_{Y, \boldsymbol{X}^t \mid \boldsymbol{x}^{1:t-1}}\left[\log p(Y, \boldsymbol{X}^t \mid \boldsymbol{x}^{1:t-1})\right] \tag{D.3}$$

$$= -\mathbb{E}_{\mathcal{M} \mid \boldsymbol{x}^{1:t-1}}\left[\mathbb{E}_{Y, \boldsymbol{X}^t \mid \mathcal{M}}\left[\log p(Y, \boldsymbol{X}^t \mid \boldsymbol{x}^{1:t-1})\right]\right] \tag{D.4}$$

$$= -\mathbb{E}_{\mathcal{M} \mid \boldsymbol{x}^{1:t-1}}\left[\mathbb{E}_{Y, \boldsymbol{X}^t \mid \mathcal{M}}\left[\log \mathbb{E}_{\mathcal{M}' \mid \boldsymbol{x}^{1:t-1}}\left[p(Y \mid \mathcal{M}') \cdot p(\boldsymbol{X}^t \mid \mathcal{M}')\right]\right]\right] \tag{D.5}$$

for any query such that query and experiment outcome are conditionally independent given an SCM. This holds true, e.g., whenever $Y$ is a deterministic function of $\mathcal{M}$ such as $Y = q_{\mathrm{CD}}(\mathcal{M}) = G$.

The marginal entropy of the experiment outcome given previously observed data is

$$H(\boldsymbol{X}^t \mid \boldsymbol{x}^{1:t-1}) = -\mathbb{E}_{\boldsymbol{X}^t \mid \boldsymbol{x}^{1:t-1}}\left[\log p(\boldsymbol{X}^t \mid \boldsymbol{x}^{1:t-1})\right] \tag{D.6}$$

$$= -\mathbb{E}_{\mathcal{M} \mid \boldsymbol{x}^{1:t-1}}\left[\mathbb{E}_{\boldsymbol{X}^t \mid \mathcal{M}}\left[\log p(\boldsymbol{X}^t \mid \boldsymbol{x}^{1:t-1})\right]\right] \tag{D.7}$$

$$= -\mathbb{E}_{\mathcal{M} \mid \boldsymbol{x}^{1:t-1}}\left[\mathbb{E}_{\boldsymbol{X}^t \mid \mathcal{M}}\left[\log \mathbb{E}_{\mathcal{M}' \mid \boldsymbol{x}^{1:t-1}}\left[p(\boldsymbol{X}^t \mid \mathcal{M}')\right]\right]\right] \tag{D.8}$$

$$= -\mathbb{E}_{\mathcal{M} \mid \boldsymbol{x}^{1:t-1}}\left[\mathbb{E}_{\boldsymbol{X}^t \mid \mathcal{M}}\left[\log \mathbb{E}_{\boldsymbol{f}', \boldsymbol{\sigma}^{2'}, G' \mid \boldsymbol{x}^{1:t-1}}\left[p(\boldsymbol{X}^t \mid \boldsymbol{f}', \boldsymbol{\sigma}^{2'}, G')\right]\right]\right] \tag{D.9}$$

$$= -\mathbb{E}_{\mathcal{M} \mid \boldsymbol{x}^{1:t-1}}\left[\mathbb{E}_{\boldsymbol{X}^t \mid \mathcal{M}}\left[\log \mathbb{E}_{G' \mid \boldsymbol{x}^{1:t-1}}\left[p(\boldsymbol{X}^t \mid G', \boldsymbol{x}^{1:t-1})\right]\right]\right] \tag{D.10}$$

$$= -\mathbb{E}_{\boldsymbol{f}, \boldsymbol{\sigma}^2, G \mid \boldsymbol{x}^{1:t-1}}\left[\mathbb{E}_{\boldsymbol{X}^t \mid \boldsymbol{f}, \boldsymbol{\sigma}^2, G}\left[\log \mathbb{E}_{G' \mid \boldsymbol{x}^{1:t-1}}\left[p(\boldsymbol{X}^t \mid G', \boldsymbol{x}^{1:t-1})\right]\right]\right] \tag{D.11}$$

$$= -\mathbb{E}_{G \mid \boldsymbol{x}^{1:t-1}}\left[\mathbb{E}_{\boldsymbol{X}^t \mid G, \boldsymbol{x}^{1:t-1}}\left[\log \mathbb{E}_{G' \mid \boldsymbol{x}^{1:t-1}}\left[p(\boldsymbol{X}^t \mid G', \boldsymbol{x}^{1:t-1})\right]\right]\right] \tag{D.12}$$

Finally, since the query posterior entropy $H(Y \mid \boldsymbol{x}^{1:t-1})$ does not depend on the candidate experiment $a_t$, we obtain

$$\underset{a_t}{\arg\max}\quad \mathrm{I}(Y; \boldsymbol{X}^t \mid \boldsymbol{x}^{1:t-1})$$

$$= \underset{a_t}{\arg\max}\quad H(Y \mid \boldsymbol{x}^{1:t-1}) + H(\boldsymbol{X}^t \mid \boldsymbol{x}^{1:t-1}) - H(Y, \boldsymbol{X}^t \mid \boldsymbol{x}^{1:t-1})$$

$$= \underset{a_t}{\arg\max}\quad H(\boldsymbol{X}^t \mid \boldsymbol{x}^{1:t-1}) - H(Y, \boldsymbol{X}^t \mid \boldsymbol{x}^{1:t-1}) \tag{D.13}$$

which, together with Eqs. (D.5) and (D.8), completes the proof. □

## D.2 Derivation of the Causal Discovery Utility Function

To derive $U_{\mathrm{CD}}(a)$, we note that $Y = q_{\mathrm{CD}}(\mathcal{M}) = G$, and hence the joint entropy of experiment outcome and query in Eq. (D.3) becomes

$$H(G, \boldsymbol{X}^t \mid \boldsymbol{x}^{1:t-1}) = -\mathbb{E}_{G, \boldsymbol{X}^t \mid \boldsymbol{x}^{1:t-1}}\left[\log p(G, \boldsymbol{X}^t \mid \boldsymbol{x}^{1:t-1})\right] \tag{D.14}$$

$$= -\mathbb{E}_{G, \boldsymbol{X}^t \mid \boldsymbol{x}^{1:t-1}}\left[\log p(\boldsymbol{X}^t \mid G, \boldsymbol{x}^{1:t-1}) + \log p(G \mid \boldsymbol{x}^{1:t-1})\right] \tag{D.15}$$

$$= -\mathbb{E}_{G, \boldsymbol{X}^t \mid \boldsymbol{x}^{1:t-1}}\left[\log p(\boldsymbol{X}^t \mid G, \boldsymbol{x}^{1:t-1})\right] + H(G \mid \boldsymbol{x}^{1:t-1}) \tag{D.16}$$

$$= -\mathbb{E}_{G \mid \boldsymbol{x}^{1:t-1}}\left[\mathbb{E}_{\boldsymbol{X}^t \mid G, \boldsymbol{x}^{1:t-1}}\left[\log p(\boldsymbol{X}^t \mid G, \boldsymbol{x}^{1:t-1})\right]\right] + H(G \mid \boldsymbol{x}^{1:t-1}). \tag{D.17}$$

Substituting this into Eq. (D.2) yields

$$I(G; \boldsymbol{X}^t \,|\, \boldsymbol{x}^{1:t-1}) \tag{D.18}$$
$$= H(\boldsymbol{X}^t \,|\, \boldsymbol{x}^{1:t-1}) + \mathbb{E}_{G \,|\, \boldsymbol{x}^{1:t-1}} \left[ \mathbb{E}_{\boldsymbol{X}^t \,|\, G, \boldsymbol{x}^{1:t-1}} \left[ \log p(\boldsymbol{X}^t \,|\, G, \boldsymbol{x}^{1:t-1}) \right] \right]. \tag{D.19}$$

By Eq. (D.12), we have

$$= \mathbb{E}_{G \,|\, \boldsymbol{x}^{1:t-1}} \left[ \mathbb{E}_{\boldsymbol{X}^t \,|\, G, \boldsymbol{x}^{1:t-1}} \left[ \log p(\boldsymbol{X}^t \,|\, G, \boldsymbol{x}^{1:t-1}) - \log \mathbb{E}_{G' \,|\, \boldsymbol{x}^{1:t-1}} \left[ p(\boldsymbol{X}^t \,|\, G', \boldsymbol{x}^{1:t-1}) \right] \right] \right] \tag{D.20}$$

which recovers the utility function $U_{\text{CD}}(a)$ from Eq. (A.13).

## D.3 Derivation of the Causal Model Learning Utility Function

To derive $U_{\text{CML}}(a)$ given $Y = q_{\text{CML}}(\mathcal{M}) = \mathcal{M}$, the joint entropy of experiment outcome and query in Eq. (D.3) are given by

$$H(\mathcal{M}, \boldsymbol{X}^t \,|\, \boldsymbol{x}^{1:t-1}) = -\mathbb{E}_{\mathcal{M}, \boldsymbol{X}^t \,|\, \boldsymbol{x}^{1:t-1}} \left[ \log p(\mathcal{M}, \boldsymbol{X}^t \,|\, \boldsymbol{x}^{1:t-1}) \right] \tag{D.21}$$
$$= -\mathbb{E}_{\mathcal{M}, \boldsymbol{X}^t \,|\, \boldsymbol{x}^{1:t-1}} \left[ \log p(\boldsymbol{X}^t \,|\, \mathcal{M}, \boldsymbol{x}^{1:t-1}) + \log p(\mathcal{M} \,|\, \boldsymbol{x}^{1:t-1}) \right] \tag{D.22}$$
$$= -\mathbb{E}_{\mathcal{M}, \boldsymbol{X}^t \,|\, \boldsymbol{x}^{1:t-1}} \left[ \log p(\boldsymbol{X}^t \,|\, \mathcal{M}) \right] + H(\mathcal{M} \,|\, \boldsymbol{x}^{1:t-1}) \tag{D.23}$$
$$= -\mathbb{E}_{\mathcal{M} \,|\, \boldsymbol{x}^{1:t-1}} \left[ \mathbb{E}_{\boldsymbol{X}^t \,|\, \mathcal{M}} \left[ \log p(\boldsymbol{X}^t \,|\, \mathcal{M}) \right] \right] + H(\mathcal{M} \,|\, \boldsymbol{x}^{1:t-1}). \tag{D.24}$$

As previously, substituting this into Eq. (D.2) yields

$$I(G; \boldsymbol{X}^t \,|\, \boldsymbol{x}^{1:t-1}) = H(\boldsymbol{X}^t \,|\, \boldsymbol{x}^{1:t-1}) + \mathbb{E}_{\mathcal{M} \,|\, \boldsymbol{x}^{1:t-1}} \left[ \mathbb{E}_{\boldsymbol{X}^t \,|\, \mathcal{M}} \left[ \log p(\boldsymbol{X}^t \,|\, \mathcal{M},) \right] \right] \tag{D.25}$$

and by Eq. (D.10), we have

$$= \mathbb{E}_{\mathcal{M} \,|\, \boldsymbol{x}^{1:t-1}} \left[ \mathbb{E}_{\boldsymbol{X}^t \,|\, \mathcal{M}} \left[ \log p(\boldsymbol{X}^t \,|\, \mathcal{M}) - \log \mathbb{E}_{G' \,|\, \boldsymbol{x}^{1:t-1}} \left[ p(\boldsymbol{X}^t \,|\, G', \boldsymbol{x}^{1:t-1}) \right] \right] \right] \tag{D.26}$$

which recovers the utility $U_{\text{CML}}(a)$ from Eq. (A.14).

For our concrete modeling choices we can further simplify this utility. Let $\mathbf{Anc}_i^{\mathcal{M}}$ and $\mathbf{Pa}_i^{\mathcal{M}}$ denote the ancestor and parent sets of node $X_i$ in $\mathcal{M}$. Then,

$$\mathbb{E}_{\mathcal{M} \,|\, \boldsymbol{x}^{1:t-1}} \left[ \mathbb{E}_{\boldsymbol{X}^t \,|\, \mathcal{M}} \left[ \log p(\boldsymbol{X}^t \,|\, \mathcal{M}) \right] \right] \tag{D.27}$$

$$= \mathbb{E}_{\mathcal{M} \,|\, \boldsymbol{x}^{1:t-1}} \left[ \mathbb{E}_{\boldsymbol{X}^t \,|\, \mathcal{M}} \left[ \log \prod_{i \notin \mathcal{I}^t} p^{\text{do}(a_t)}(\boldsymbol{X}_i^t \,|\, \mathbf{pa}_i^{\mathcal{M}}, \mathcal{M}) \right] \right] \tag{D.28}$$

$$= \mathbb{E}_{\mathcal{M} \,|\, \boldsymbol{x}^{1:t-1}} \left[ \mathbb{E}_{\boldsymbol{X}^t \,|\, \mathcal{M}} \left[ \sum_{i \notin \mathcal{I}^t} \log p^{\text{do}(a_t)}(\boldsymbol{X}_i^t \,|\, \mathbf{pa}_i^{\mathcal{M}}, \mathcal{M}) \right] \right] \tag{D.29}$$

$$= \mathbb{E}_{\mathcal{M} \,|\, \boldsymbol{x}^{1:t-1}} \left[ \sum_{i \notin \mathcal{I}^t} \mathbb{E}_{\boldsymbol{X}^t \,|\, \mathcal{M}} \left[ \log p^{\text{do}(a_t)}(\boldsymbol{X}_i^t \,|\, \mathbf{pa}_i^{\mathcal{M}}, \mathcal{M}) \right] \right] \tag{D.30}$$

$$= \mathbb{E}_{\mathcal{M} \,|\, \boldsymbol{x}^{1:t-1}} \left[ \sum_{i \notin \mathcal{I}^t} \mathbb{E}_{\mathbf{Anc}_i^{\mathcal{M}} \,|\, \text{do}(a_t), \mathcal{M}} \left[ \mathbb{E}_{\boldsymbol{X}_i^t \,|\, \mathbf{pa}_i^{\mathcal{M}}, \text{do}(a_t), \mathcal{M}} \left[ \log p^{\text{do}(a_t)}(\boldsymbol{X}_i^t \,|\, \mathbf{pa}_i^{\mathcal{M}}, \mathcal{M}) \right] \right] \right]. \tag{D.31}$$

Since our root nodes and GPs assume an additive Gaussian noise model, the innermost expectation amounts to the negative entropy the Gaussian noise variable, i.e.,

$$\mathbb{E}_{\boldsymbol{X}_i^t \,|\, \mathbf{pa}_i^{\mathcal{M}}, \text{do}(a_t), \mathcal{M}} \left[ \log p^{\text{do}(a_t)}(\boldsymbol{X}_i^t \,|\, \mathbf{pa}_i^{\mathcal{M}}, \mathcal{M}) \right] = -\frac{N_t}{2} \log(2\pi\sigma_i^2 e). \tag{D.32}$$

As we further assume a homoscedastic noise model for our GPs, Eq. (D.31) reduces to

$$\mathbb{E}_{\mathcal{M}\,|\,\boldsymbol{x}^{1:t-1}} \left[ \sum_{i\notin\mathcal{I}^t} -\frac{N_t}{2} \log(2\pi\sigma_i^2 e) \right] \tag{D.33}$$

$$= -\mathbb{E}_{\boldsymbol{f},\boldsymbol{\sigma}^2,G\,|\,\boldsymbol{x}^{1:t-1}} \left[ \sum_{i\notin\mathcal{I}^t} \frac{N_t}{2} \log(2\pi\sigma_i^2 e) \right] \tag{D.34}$$

$$= -\mathbb{E}_{G\,|\,\boldsymbol{x}^{1:t-1}} \left[ \mathbb{E}_{\boldsymbol{\sigma}^2\,|\,G,\boldsymbol{x}^{1:t-1}} \left[ \sum_{i\notin\mathcal{I}^t} \frac{N_t}{2} \log(2\pi\sigma_i^2 e) \right] \right] \tag{D.35}$$

$$= -\mathbb{E}_{G\,|\,\boldsymbol{x}^{1:t-1}} \left[ \sum_{i\notin\mathcal{I}^t} \mathbb{E}_{\sigma_i^2\,|\,G,\boldsymbol{x}^{1:t-1}} \left[ \frac{N_t}{2} \log(2\pi\sigma_i^2 e) \right] \right], \tag{D.36}$$

which can be approximated by nested Monte Carlo estimation. For non-root nodes we approximate the inner expectation with a single point estimate (cf. Section A.1). For root nodes we can compute the inner expectation in closed form as

$$\mathbb{E}_{\sigma_i^2\,|\,G,\boldsymbol{x}^{1:t-1}} \left[ \frac{N_t}{2} \log(2\pi\sigma_i^2 e) \right] = \frac{N_t}{2} \left( \log(2\pi e) - \psi(\alpha_i^t) + \log \beta_i^t \right) \tag{D.37}$$

where $\alpha_i^t, \beta_i^t$ are the parameters of the inverse-gamma noise posterior $\sigma_i^2 \sim \Gamma^{-1}(\sigma_i^2\,|\,\alpha_i^t, \beta_i^t)$ (see Appx. E.3) and $\psi(\cdot)$ is the digamma function.

**Proof** (adapted from [61]). We need to show that

$$\mathbb{E}_{\sigma^2} \left[ \log(\sigma^2) \right] = -\psi(\alpha) + \log \beta \tag{D.38}$$

where the noise variance $\sigma^2$ follows an inverse-gamma density

$$\sigma^2 \sim \Gamma^{-1}(\sigma^2\,|\,\alpha, \beta) = \frac{\beta^\alpha}{\Gamma(\alpha)} \cdot (\sigma^2)^{-\alpha-1} \cdot e^{-\frac{\beta}{\sigma^2}}. \tag{D.39}$$

By substituting $y = \log \sigma^2$ we get

$$y \sim p(y\,|\,\alpha, \beta) = \frac{\beta^\alpha}{\Gamma(\alpha)} \cdot e^{-\alpha y} \cdot e^{-\beta e^{-y}}. \tag{D.40}$$

Now note that

$$\int_{-\infty}^{\infty} p(y\,|\,\alpha, \beta) dy = 1 \tag{D.41}$$

and hence

$$\frac{\Gamma(\alpha)}{\beta^\alpha} = \int_{-\infty}^{\infty} e^{-\alpha y} \cdot e^{-\beta e^{-y}} dy. \tag{D.42}$$

By differentiating the latter integrand w.r.t. $\alpha$ we get

$$\frac{d}{d\alpha} \left( e^{-\alpha y} \cdot e^{-\beta e^{-y}} \right) = (-y)e^{-\alpha y} \cdot e^{-\beta e^{-y}} = (-y) \cdot p(y\,|\,\alpha, \beta) \cdot \frac{\Gamma(\alpha)}{\beta^\alpha}. \tag{D.43}$$

Bringing the parts together we obtain

$$\mathbb{E}_{\sigma^2}\left[\log(\sigma^2)\right] = \qquad\qquad \mathbb{E}_y\left[y\right] \tag{D.44}$$

$$= \qquad \int_{-\infty}^{\infty} y \cdot p(y \,|\, \alpha, \beta) dy \tag{D.45}$$

$$\overset{Eq. \text{ (D.43)}}{=} \quad -\frac{\beta^\alpha}{\Gamma(\alpha)} \int_{-\infty}^{\infty} \frac{d}{d\alpha}\left(e^{-\alpha y} \cdot e^{-\beta e^{-y}}\right) dy \tag{D.46}$$

$$= \quad -\frac{\beta^\alpha}{\Gamma(\alpha)} \frac{d}{d\alpha}\left(\int_{-\infty}^{\infty} e^{-\alpha y} \cdot e^{-\beta e^{-y}} dy\right) \tag{D.47}$$

$$\overset{Eq. \text{ (D.42)}}{=} \quad -\frac{\beta^\alpha}{\Gamma(\alpha)} \frac{d}{d\alpha}\left(\frac{\Gamma(\alpha)}{\beta^\alpha}\right) \tag{D.48}$$

$$= \quad -\frac{\beta^\alpha}{\Gamma(\alpha)}\left(\beta^{-\alpha} \cdot \frac{d}{d\alpha}\Gamma(\alpha) - \Gamma(\alpha) \cdot \beta^{-\alpha} \cdot \log\beta\right) \tag{D.49}$$

$$= \quad -\psi(\alpha) + \log\beta, \tag{D.50}$$

which completes the proof. $\qquad\square$

In summary, in our instance of GP-DIBS-ABCI we estimate the causal model learning utility as

$$U_{\text{CML}}(a_t) = -\mathbb{E}_{G \,|\, \boldsymbol{x}^{1:t-1}}\Bigg[ \sum_{i \in \mathbf{R}(G)\setminus\mathcal{I}^t} \frac{N_t}{2}\left(\log(2\pi e) - \psi(\alpha_i^t) + \log\beta_i^t\right) +$$

$$\sum_{i \in \mathbf{NR}(G)\setminus\mathcal{I}^t} \mathbb{E}_{\sigma_i^2 \,|\, G,\boldsymbol{x}^{1:t-1}}\left[\frac{N_t}{2}\log(2\pi\sigma_i^2 e)\right] +$$

$$\mathbb{E}_{\boldsymbol{X}^t \,|\, G,\boldsymbol{x}^{1:t-1}}\left[\log\mathbb{E}_{G' \,|\, \boldsymbol{x}^{1:t-1}}\left[p(\boldsymbol{X}^t \,|\, G', \boldsymbol{x}^{1:t-1})\right]\right]\Bigg] \tag{D.51}$$

### D.4 Derivation of the Causal Reasoning Utility Function

We derive the utility function $U_{\text{CR}}(a)$ in Eq. (A.12) for the query $Y = X_j^{\text{do}(X_i=\psi)}$ with $\psi \sim p(\psi)$ a distribution over intervention values. Starting with the joint entropy in Eq. (D.3) we marginalise over graphs (instead of SCMs) to exploit that we can sample from and evaluate $p(\boldsymbol{X} \,|\, G, \boldsymbol{x}^{1:t-1})$ in closed form by using GPs:

$$-H(Y, \boldsymbol{X}^t \,|\, \boldsymbol{x}^{1:t-1})$$

$$= \mathbb{E}_{Y, \boldsymbol{X}^t \,|\, \boldsymbol{x}^{1:t-1}}\left[\log p(Y, \boldsymbol{X}^t \,|\, \boldsymbol{x}^{1:t-1})\right] \tag{D.52}$$

$$= \mathbb{E}_{G \,|\, \boldsymbol{x}^{1:t-1}}\left[\mathbb{E}_{Y,\boldsymbol{X}^t \,|\, G,\boldsymbol{x}^{1:t-1}}\left[\log\mathbb{E}_{G' \,|\, \boldsymbol{x}^{1:t-1}}\left[p(Y, \boldsymbol{X}^t \,|\, G', \boldsymbol{x}^{1:t-1})\right]\right]\right] \tag{D.53}$$

$$= \mathbb{E}_{G \,|\, \boldsymbol{x}^{1:t-1}}\Big[\mathbb{E}_{\boldsymbol{X}^t \,|\, G,\boldsymbol{x}^{1:t-1}}\Big[\mathbb{E}_{Y \,|\, \boldsymbol{X}^t,G,\boldsymbol{x}^{1:t-1}}\Big[$$

$$\log\mathbb{E}_{G' \,|\, \boldsymbol{x}^{1:t-1}}\left[p(Y \,|\, \boldsymbol{X}^t, G', \boldsymbol{x}^{1:t-1}) \cdot p(\boldsymbol{X}^t \,|\, G', \boldsymbol{x}^{1:t-1})\right]\Big]\Big]\Big] \tag{D.54}$$

To estimate $\mathbb{E}_{Y \,|\, \boldsymbol{X}^t,G,\boldsymbol{x}^{1:t-1}}[\cdot]$ we first sample intervention values $\psi \sim p(\psi)$ and then sample from the respective interventional densities $p^{\text{do}(X_i=\psi)}(X_j \,|\, \boldsymbol{X}^t, G, \boldsymbol{x}^{1:t-1})$ induced by candidate SCMs with graph $G$. Thus, the expectation becomes $\mathbb{E}_\psi\left[\mathbb{E}_{X_j \,|\, \boldsymbol{X}^t,G,\boldsymbol{x}^{1:t-1}}^{\text{do}(X_i=\psi)}[\cdot]\right]$. To evaluate $p(Y \,|\, \boldsymbol{X}^t, G', \boldsymbol{x}^{1:t-1})$ we estimate $p^{\text{do}(X_i=\psi)}(X_j \,|\, \boldsymbol{X}^t, G', \boldsymbol{x}^{1:t-1})$ as described in Appx. E.1. The joint entropy therefore becomes

$$-H(Y, \boldsymbol{X}^t \,|\, \boldsymbol{x}^{1:t-1}) = \mathbb{E}_{G \,|\, \boldsymbol{x}^{1:t-1}}\Big[\mathbb{E}_{\boldsymbol{X}^t \,|\, G,\boldsymbol{x}^{1:t-1}}\Big[\mathbb{E}_\psi\Big[\mathbb{E}_{X_j \,|\, \boldsymbol{X}^t,G,\boldsymbol{x}^{1:t-1}}^{\text{do}(X_i=\psi)}\Big[ \tag{D.55}$$

$$\log\mathbb{E}_{G' \,|\, \boldsymbol{x}^{1:t-1}}\left[p^{\text{do}(X_i=\psi)}(X_j \,|\, \boldsymbol{X}^t, \boldsymbol{x}^{1:t-1}) \cdot p(\boldsymbol{X}^t \,|\, G', \boldsymbol{x}^{1:t-1})\right]\Big]\Big]\Big]\Big]$$

By substituting Eqs. (D.13) and (E.1) into Eq. (D.12) we obtain the causal reasoning utility in Eq. (A.12).

# E  Approximate Inference and Experimental Details

In this section, we provide details about our approximate inference and estimation procedures, including the estimation of the marginal interventional likelihoods in Section E.1 and prior choices in Sections E.2 — E.4. We also provide details on DiBS for approximate graph posterior inference in Section E.5, the estimation of the information gain utilities in Section E.6, and our use of Bayesian Optimisation for experimental design in Section E.7.

## E.1  Estimating Posterior Marginal Interventional Likelihoods

In the following, we show how we estimate (posterior) marginal interventional likelihoods $p^{\mathrm{do}(x_j)}(\boldsymbol{x}_i \,|\, \boldsymbol{x}^{1:t})$. Let $\mathbf{Anc}_i^G$ and $\mathbf{Pa}_i^G$ denote the ancestor and parent sets of node $X_i$ in $G$. Then, the marginal interventional likelihood is given by

$$p^{\mathrm{do}(x_j)}(\boldsymbol{x}_i \,|\, \boldsymbol{x}^{1:t})$$

$$= \mathbb{E}_{\mathcal{M} \,|\, \boldsymbol{x}^{1:t}} \left[ p^{\mathrm{do}(x_j)}(\boldsymbol{x}_i \,|\, \mathcal{M}) \right] \tag{E.1}$$

$$= \mathbb{E}_{\boldsymbol{f},\boldsymbol{\sigma}^2,G \,|\, \boldsymbol{x}^{1:t}} \left[ p^{\mathrm{do}(x_j)}(\boldsymbol{x}_i \,|\, \boldsymbol{f},\boldsymbol{\sigma}^2,G) \right] \tag{E.2}$$

$$= \mathbb{E}_{\boldsymbol{f},\boldsymbol{\sigma}^2,G \,|\, \boldsymbol{x}^{1:t}} \left[ \mathbb{E}_{\mathbf{Anc}_i^G \,|\, \mathrm{do}(x_j),\boldsymbol{f},\boldsymbol{\sigma}^2,G} \left[ p^{\mathrm{do}(x_j)}(\boldsymbol{x}_i \,|\, \mathbf{anc}_i^G,\boldsymbol{f},\boldsymbol{\sigma}^2,G) \right] \right]. \tag{E.3}$$

Given that $X_i$ is independent of it's non-descendants given its parents, we obtain

$$= \mathbb{E}_{\boldsymbol{f},\boldsymbol{\sigma}^2,G \,|\, \boldsymbol{x}^{1:t}} \left[ \mathbb{E}_{\mathbf{Anc}_i^G \,|\, \mathrm{do}(x_j),\boldsymbol{f},\boldsymbol{\sigma}^2,G} \left[ p^{\mathrm{do}(x_j)}(\boldsymbol{x}_i \,|\, \mathbf{pa}_i^G, f_i, \sigma_i^2, G) \right] \right] \tag{E.4}$$

$$= \mathbb{E}_{G \,|\, \boldsymbol{x}^{1:t}} \left[ \mathbb{E}_{\boldsymbol{f},\boldsymbol{\sigma}^2 \,|\, G,\boldsymbol{x}^{1:t}} \left[ \mathbb{E}_{\mathbf{Anc}_i^G \,|\, \mathrm{do}(x_j),\boldsymbol{f},\boldsymbol{\sigma}^2,G} \left[ p^{\mathrm{do}(x_j)}(\boldsymbol{x}_i \,|\, \mathbf{pa}_i^G, f_i, \sigma_i^2, G) \right] \right] \right]. \tag{E.5}$$

Given that $p(\boldsymbol{f},\boldsymbol{\sigma}^2 \,|\, G, \boldsymbol{x}^{1:t})$ factorises and $\mathbf{Anc}_i^G$ are independent of mechanisms and noise variances $\boldsymbol{f},\boldsymbol{\sigma}^2$ of the non-ancestors of $X_i$, we have

$$= \mathbb{E}_{G \,|\, \boldsymbol{x}^{1:t}} \left[ \mathbb{E}_{\boldsymbol{f}_{\mathbf{Anc}_i^G},\boldsymbol{\sigma}^2_{\mathbf{Anc}_i^G} \,|\, G,\boldsymbol{x}^{1:t}} \left[ \mathbb{E}_{\mathbf{Anc}_i^G \,|\, \mathrm{do}(x_j),\boldsymbol{f}_{\mathbf{Anc}_i^G},\boldsymbol{\sigma}^2_{\mathbf{Anc}_i^G},G} \Big[ \right. \right.$$
$$\left. \left. \mathbb{E}_{f_i,\sigma_i^2 \,|\, G,\boldsymbol{x}^{1:t}} \left[ p^{\mathrm{do}(x_j)}(\boldsymbol{x}_i \,|\, \mathbf{pa}_i^G, f_i, \sigma_i^2, G) \right] \Big] \right] \right]. \tag{E.6}$$

Finally, marginalising out the functions and noise variances, we obtain

$$= \mathbb{E}_{G \,|\, \boldsymbol{x}^{1:t}} \left[ \mathbb{E}_{\boldsymbol{f}_{\mathbf{Anc}_i^G},\boldsymbol{\sigma}^2_{\mathbf{Anc}_i^G} \,|\, G,\boldsymbol{x}^{1:t}} \left[ \mathbb{E}_{\mathbf{Anc}_i^G \,|\, \mathrm{do}(x_j),\boldsymbol{f}_{\mathbf{Anc}_i^G},\boldsymbol{\sigma}^2_{\mathbf{Anc}_i^G},G} \left[ p^{\mathrm{do}(x_j)}(\boldsymbol{x}_i \,|\, \mathbf{pa}_i^G, G) \right] \right] \right] \tag{E.7}$$

$$= \mathbb{E}_{G \,|\, \boldsymbol{x}^{1:t}} \left[ \mathbb{E}_{\mathbf{Anc}_i^G \,|\, \mathrm{do}(x_j),G} \left[ p^{\mathrm{do}(x_j)}(\boldsymbol{x}_i \,|\, \mathbf{pa}_i^G, G) \right] \right] \tag{E.8}$$

$$= \mathbb{E}_{G \,|\, \boldsymbol{x}^{1:t}} \left[ \mathbb{E}_{\mathbf{Anc}_i^G \,|\, \mathrm{do}(x_j),G} \left[ p(\boldsymbol{x}_i \,|\, \mathbf{pa}_i^G, G) \Big|_{X_j=x_j} \right] \right]. \tag{E.9}$$

We use Monte Carlo estimation to approximate the outer expectation of this quantity according to Eq. (A.10). To approximate the inner expectation by performing ancestral sampling from the interventional density $p^{\mathrm{do}(x_j)}(\boldsymbol{X} \,|\, G)$, where we use 50 samples when estimating the $U_{\mathrm{CR}}$ utility in Equation Eq. (A.12) and 200 samples when estimating the metrics described in Appx. G.

## E.2  Sampling Ground Truth Graphs

When generating ground truth SCMs for evaluation, we sample causal graphs according to two random graph models. First, we sample scale-free graphs using the preferential attachment process presented by Barabási and Albert [5]. We use the `networkx.generators.barabasi_albert_graph` implementation provided in the NetworkX [22] Python package and interpret the returned, undirected graph as a DAG by only considering the upper-triangular part of its adjacency matrix. Before permuting the node labels, we generate graphs with in-degree 2 for nodes $\{X_i\}_{i=3}^d$ whereas $X_1$ and $X_2$ are always

root nodes. In addition, we consider Erdös-Renyi random graphs [15], where edges are sampled independently with probability $p = \frac{4}{d-1}$. After sampling edges, we choose a random ordering and discard any edges that disobey this ordering to obtain a DAG. Our choice of $p$ yields an expected degree of 2. Unlike Lorch et al. [35], we do not provide our model with any kind of prior information on the graph structure.

## E.3 Normal-Inverse-Gamma Prior for Root Nodes

We use a conjugate normal-inverse-gamma (N-$\Gamma^{-1}$) prior

$$p(f_i, \sigma_i^2 \mid G) = \text{N-}\Gamma^{-1}(\mu_i, \lambda_i, \alpha_i^R, \beta_i^R) \tag{E.10}$$

as the joint prior over functions and noise parameters for root nodes in $G$ (see Section A and Fig. 4). In our experiments, we use $\mu_i = 0$, $\lambda_i = 0.1$, $\alpha_i^R = 50$ and $\beta_i^R = 25$. When generating ground truth SCMs, we draw one sample for $(f_i^\star, \sigma_i^{2,\star})$ from this prior for all $i$ and leave it fixed thereafter. Closed-form expressions for the (posterior) marginal likelihood can be found, e.g., in [41].

## E.4 Gamma Priors for GP Hyperparameters of Non-Root Nodes

We model non-root node mechanisms with GPs (see Section A.1), where each GP has a set of hyperparameters $(\boldsymbol{\kappa}_i, \sigma_i^2)$ where $\boldsymbol{\kappa}_i = (\kappa_i^l, \kappa_i^o)$ includes a length scale and output scale parameter, respectively, and where $\sigma_i^2$ denotes the variance of the Gaussian noise variable $U_i$. In our experiments, we use $p(\sigma_i^2 \mid G) = \text{Gamma}(\alpha = 50, \beta = 500)$, $p(\kappa_i^o \mid G) = \text{Gamma}(\alpha = 100, \beta = 10)$ and $p(\kappa_i^l \mid G) = \text{Gamma}(\alpha = 30 \cdot |\mathbf{Pa}_i^G|, \beta = 30)$, where $|\mathbf{Pa}_i^G|$ denotes the size of the parent set of $X_i$ in $G$.

## E.5 DiBS for Approximate Posterior Graph Inference

DiBS [35] introduces a probabilistic latent space representation for DAGs to allow for efficient posterior inference in continuous space. Specifically, given some latent particle $\boldsymbol{z} \in \mathbb{R}^{d \times d \times 2}$ we can define an edge-wise generative model

$$p(G \mid \boldsymbol{z}) = \prod_{i=1}^{d} \prod_{\substack{j=1 \\ j \neq i}}^{d} p(G_{i,j} \mid \boldsymbol{z}) \tag{E.11}$$

where $G_{i,j} \in \{0, 1\}$ indicates the absence/presence of an edge from $X_i$ to $X_j$ in $G$, and a prior distribution

$$p(\boldsymbol{Z}) \propto \exp(-\beta \, \mathbb{E}_{G \mid \boldsymbol{Z}} [h(G)]) \prod_{i,j,k} \mathcal{N}(z_{i,j,k} \mid 0, 1) \tag{E.12}$$

where $h(G)$ is a scoring function quantifying the "degree of cyclicity" of $G$. $\beta$ is a temperature parameter weighting the influence of the expected cyclicity in the prior. Lorch et al. [35] propose to use Stein Variational Gradient Descent [34] for approximate inference of $p(\boldsymbol{Z} \mid \boldsymbol{x}^{1:t})$. SVGD maintains a fixed set of particles $\boldsymbol{z} = \{\boldsymbol{z}_m\}_{m=1}^M$ and updates them using the posterior score $\nabla \log p(\boldsymbol{z} \mid \boldsymbol{x}^{1:t}) = \nabla \log p(\boldsymbol{z}) + \nabla \log p(\boldsymbol{x}^{1:t} \mid \boldsymbol{z})$. In our experiments, we use $K = 5$ latent particles. For the estimation of expectations as in Eq. (A.10), we use $K = 40$ MC graph samples unless otherwise stated, hence, a total of $M \cdot K = 200$ graphs, and we use the DiBS+ particle weighting. In contrast to the original DiBS version, we do not use the annealing parameter $\alpha$ to force the mass of $p(G|\boldsymbol{z})$ onto a single graph during training. For further details on the method and its implementation, we refer to the original publication [35] and the provided code.

## E.6 Estimation of the Information Gain Utility Functions

When estimating the information gain utilities (see Appxs. A.2 and D), we keep the set of Monte Carlo samples from the SCM posterior $p(\mathcal{M} \mid \boldsymbol{x}^{1:t})$ fixed for all evaluations of the chosen utility during a given experiment design phase at time $t$, i.e., during the optimisation for all candidate intervention sets and intervention targets. In our experiments, for the $U_{\text{CD}}$ and $U_{\text{CML}}$ utilities we sample 5 and 30 graphs to approximate the outer and inner expectations w.r.t. the posterior graphs, respectively.

We sample 100 hypothetical experiment outcomes with given batch size from $p(\boldsymbol{X}^t \,|\, G, \boldsymbol{x}^{1:t})$ to approximate the expectation $\mathbb{E}_{\boldsymbol{X}^t \,|\, G, \boldsymbol{x}^{1:t}} [\cdot]$.

For the $U_{\mathrm{CR}}$ utility we sample 3 and 9 graphs to approximate the outer and inner expectations w.r.t. the posterior graphs, respectively. We sample 50 hypothetical experiment outcomes with given batch size from $p(\boldsymbol{X}^t \,|\, G, \boldsymbol{x}^{1:t})$ to approximate expectations of the form $\mathbb{E}_{\boldsymbol{X}^t \,|\, G, \boldsymbol{x}^{1:t}} [\cdot]$. To approximate the expectations $\mathbb{E}_\psi \big[ \mathbb{E}^{\mathrm{do}(X_i=\psi)}_{X_j \,|\, \boldsymbol{X}^t, G, \mathcal{D}} [\cdot] \big]$ we sample 5 intervention values from $p(\psi)$ and draw 3 samples from $p^{\mathrm{do}(X_i=\psi)}(X_j \,|\, \boldsymbol{X}^t, G, \mathcal{D})$ for each intervention value.

### E.7  Bayesian Optimisation for Experimental Design

In order to find the optimal experiment $a_t^\star = (\mathcal{I}^\star, \boldsymbol{x}_\mathcal{I}^\star)$ at time $t$, we compute the optimal intervention value $\boldsymbol{x}_\mathcal{I}^\star \in \arg\max_{\boldsymbol{x}} U(\mathcal{I}, \boldsymbol{x})$ for each candidate intervention target set $\mathcal{I}$ (see Eq. (A.15)). As the evaluation of our proposed utility functions $U(a)$ is expensive, we require an efficient approach for finding optimal intervention values using as few function evaluations as possible. Following von Kügelgen et al. [67], we employ *Bayesian optimisation* (BO) [36, 37] for this task and model our uncertainty in $U(\mathcal{I}, \boldsymbol{x})$ given previous evaluations $\mathcal{D}_{BO} = \{(\boldsymbol{x}_l, U(\mathcal{I}, \boldsymbol{x}_l))\}_{l=1}^k$ with a GP. We select a new candidate solution according to the GP-UCB acquisition function [63],

$$\mathbf{x}_{k+1} = \arg\max_{\mathbf{x}} \mu_k(\mathbf{x}) + \gamma \sigma_k(\mathbf{x}) , \tag{E.13}$$

where $\mu_k(\mathbf{x})$ and $\sigma_k(\mathbf{x})$ correspond to the mean and standard deviation of the GP predictive distribution $p(U(\mathcal{I}, \boldsymbol{x}) \,|\, \mathcal{D}_{BO})$ (see Appx. C). We then evaluate $U(\mathcal{I}, \boldsymbol{x}_{k+1})$ at the selected $\boldsymbol{x}_{k+1}$ and repeat. The scalar factor $\gamma$ trades off exploitation with exploration. In our experiments, we set $\gamma = 1$ and run the GP-UCB algorithm 8 times for each candidate set of intervention targets.

**Algorithm 2:** Particle Resampling

---

**Input:** set of latent particles $\boldsymbol{z} = \{\boldsymbol{z}_k\}_{k=1}^K$
**Output:** set of resampled latent particles $\tilde{\boldsymbol{z}} = \{\tilde{\boldsymbol{z}}_k\}_{k=1}^K$

$\tilde{\boldsymbol{z}} \leftarrow \varnothing$                       ▷initialise set of resampled particles
$N_{max} \leftarrow \left\lceil \frac{K}{4} \right\rceil$                      ▷max. number of particles to keep
$\{w_k\}_{k=1}^K \leftarrow \left\{ \frac{p(\boldsymbol{z}_k \,|\, \boldsymbol{x}^{1:t})\, \tilde{p}(\boldsymbol{z}_k)}{\sum_k p(\boldsymbol{z}_k \,|\, \boldsymbol{x}^{1:t})\, \tilde{p}(\boldsymbol{z}_k)} \right\}$            ▷compute particle weights
$n_{kept} \leftarrow 0$
**for** $w_k$ **in** sort_descending($\{w_k\}_{k=1}^K$) **do**
    **if** $n_{kept} < N_{max}$ **and** $w_k > 0.01$ **then**
        $\tilde{\boldsymbol{z}} \leftarrow \tilde{\boldsymbol{z}} \cup \{\boldsymbol{z}_k\}$ $n_{kept} \leftarrow n_{kept} + 1$
    **end**
    **else**
        $\boldsymbol{z}_{new} \sim p(\boldsymbol{Z})$
        $\tilde{\boldsymbol{z}} \leftarrow \tilde{\boldsymbol{z}} \cup \{\boldsymbol{z}_{new}\}$
    **end**
**end**

---

# F   Implementation Details

In this section, we give details about our implementation, including our particle resampling procedure in Section F.1, the sharing and caching of priors in Section F.2, a discussion of the computational complexity of our implementation in Section F.3, and finally some information on our code framework and computing resources in Section F.4. Our implementation is available at https://www.github.com/chritoth/active-bayesian-causal-inference.

## F.1   Particle Resampling

As described in Alg. 1, we resample latent particles $\boldsymbol{z} = \{\boldsymbol{z}_k\}_{k=1}^K$ according to a predefined schedule instead of sampling new particles from the particle prior $p(\boldsymbol{Z})$ after each epoch. Although sampling new particles would allow for higher diversity in the graph Monte Carlo samples and their respective mechanisms, it also entails a higher computational burden as the caching of mechanism marginal log-likelihoods is not as effective anymore. On the other hand, keeping a subset of the inferred particles is efficient, because once we have inferred a "good" particle $\boldsymbol{z}_k$ that supposedly has a high posterior density $p(\boldsymbol{z}_k \,|\, \boldsymbol{x}^{1:t})$ it would be wasteful to discard the particle only to infer a similar particle again. Empirically, we found that keeping particles depending on their unnormalized posterior densities according to Alg. 2 does not diminish inference quality while increasing computational efficiency. In our experiments, we chose the following resampling schedule:

$$r_t = \begin{cases} 1 & \text{if} \quad t \in \{1, 2, 3, 4, 5, 6, 9\} \\ 1 & \text{if} \quad t \bmod 5 = 0 \\ 0 & \text{otherwise.} \end{cases}$$

## F.2   Shared Priors and Caching of Marginal Likelihoods

We share priors for mechanisms and noise $p(f_i, \sigma_i \,|\, G)$, as well as for GP hyperparameters $p(\boldsymbol{\kappa}_i \,|\, G)$, across all graphs $G$ that induce the same parent set $\mathbf{Pa}_i^G$. Consequently, not only the posteriors $p(f_i, \sigma_i \,|\, G, \boldsymbol{x}^{1:t})$ and $p(\boldsymbol{\kappa}_i \,|\, G, \boldsymbol{x}^{1:t})$, but also the GP marginal likelihoods $p(\boldsymbol{x}_i^{1:t} \,|\, G)$ and GP predictive marginal likelihoods $p(\boldsymbol{x}_i^{t+1} \,|\, G, \boldsymbol{x}^{1:t})$ can be shared across graphs with identical parent sets for node $X_i$. By caching the values of the computed GP (posterior) marginal likelihoods, we substantially save on computational cost when computing expectations of the form $\mathbb{E}_{G \,|\, \boldsymbol{z}} \left[ p(\boldsymbol{x}^{1:t} \,|\, G)\, \phi(G) \right]$ and $\mathbb{E}_{G \,|\, \boldsymbol{z}} \left[ p(\boldsymbol{x}^{t+1} \,|\, G, \boldsymbol{x}^{1:t})\, \phi(G) \right]$ where $\phi(G)$ is some quantity depending the graph.

Specifically, consider that $p(\mathbf{x}^{1:t}|G) = \prod_i p(\mathbf{x}_i^{1:t}|G)$ factorizes into the GP marginal likelihoods of the individual mechanisms, so for $d$ nodes in the graph and $N$ samples in $\mathbf{x}^{1:t}$ (counted over all time steps) the complexity of computing $p(\mathbf{x}^{1:t}|G)$ is $O(d \cdot N^3)$ for a fixed set of GP hyperparameters (for

simplicity we ignore that not all $d$ mechanisms are modelled by GPs as some are root nodes, so this is not a tight bound). Thus, in the worst case, estimating $p(\mathbf{x}^{1:t}|z) = \mathbb{E}_{G|z}[p(\mathbf{x}^{1:t}|G)]$ with K graph samples would yield a complexity of $O(K \cdot d \cdot N^3)$. By caching the marginal likelihoods as outlined above we can rewrite the complexity $O(K \cdot d \cdot N^3)$ as $O(L \cdot N^3)$ where $L \leq K \cdot d$ denotes the number of unique mechanisms entailed by the set of $K$ graph samples. Although this does not reduce the worst case complexity it nevertheless greatly alleviates the computational demand in practice.

The benefit of caching becomes even more pronounced as $p(G \,|\, \boldsymbol{z})$ concentrates is mass on a small set of similar graphs as a result of the inference process. In particular, when updating the latent particles using SVGD we do not need to recompute $p(\boldsymbol{x}^{1:t} \,|\, G)$ after we have once before sampled $G$, which greatly speeds up the gradient estimation of the particle posterior.

### F.3 Computational Complexity

There are two main phases in our algorithm (disregarding the computation of metrics for evaluation), (i) the inference phase where we (approximately) infer the posterior over SCMs $p(\mathcal{M}|\mathbf{x}^{1:t})$ after collecting new experimental data, and (ii) the experimental design phase.

The inference phase has worst-case complexity in $O(T_{SVGD} \cdot (T_{HP} \cdot M \cdot K \cdot d \cdot N^3 + M^2 \cdot d^2))$ where $T_{SVGD}$ is the number of SVGD update steps, $T_{HP}$ is the number of GP hyperparameter update steps, $M$ is the number of latent $z$ particles, $K$ is the number of graph samples per latent $z$ particle, $d$ is the number of nodes in the network, and $N$ is the number of collected experimental samples in $\mathbf{x}^{1:t}$. The computation of the GP marginal likelihood dominates the complexity of the inference phase. To improve scalability we make use of shared priors and caching. Additionally, we update the GP hyperparameters according to a predefined schedule instead of doing so after each performed experiment. In our experiments, both measures reduce the factor $T_{HP} \cdot M \cdot K \cdot d$ significantly. For example, running inference with 5 freshly initialized $z$ particles with 40 graph samples each on a scale-free SCM with 20 nodes updates the hyperparameters of (2970, 964, 177) GPs during SVGD update steps (1, 5, 10), and of less than 15 GPs after 20 SVGD update steps. Compared to $M \cdot K \cdot d = 4000$ in this example, the benefit is evident.

In the experimental design phase we parallelize finding the optimal intervention value for each candidate target node, so the complexity is basically the number of Bayesian optimization (BO) steps times the complexity of the utility we want to optimize for. For a general query (cf. Eq. (A.11)) we have complexity in $O(T_{BO} \cdot M \cdot K_{outer} \cdot S \cdot Q \cdot M \cdot K_{inner} \cdot (O(p(y|\mathcal{M}) + d \cdot N^3)))$ where $T_{BO}$ is the number of Bayesian optimization iterations, $M$ is the number of latent $z$ particles, $K_{outer}$ is the number of graph samples in the outer SCM expectation in, $S$ is the number of simulated experiments per SCM, $Q$ is the number of simulated queries, $K_{inner}$ is the number of graph samples in the inner SCM expectation, $O(\,p(y|\mathcal{M})\,)$ is the complexity of evaluating the query likelihood and $d \cdot N^3$ is the complexity of evaluating the GP predictive posteriors for the simulated experiments. For the causal discovery and model learning utilities the complexity reduces to $O(T_{BO} \cdot M \cdot K_{outer} \cdot S \cdot M \cdot K_{inner} \cdot d \cdot N^3)$.

In summary, the complexity of our ABCI implementation is dominated the experimental design phase from a high-level perspective. On a lower level, the cubic scaling of GP inference is the major computational issue that we alleviate by caching the (posterior) marginal log-likelihoods (see Appx. F.2 for details). However, in a small data regime where experimental data is costly to obtain, GPs are not a prohibitive element in our inference chain. Furthermore, GP scaling issues could be alleviated, e.g., by using sparse GP approximation or any other kind of scalable Bayesian mechanism model. Disregarding issues of GP scaling, the estimation of the information gain utilities is still costly, simply because it requires many levels of nested sampling and too few Monte-Carlo samples will yield too noisy, in the worst case unusable utility estimates. We believe that in follow-up work much can be gained in terms of scalability as well as performance by incorporating recent advances in nested Monte-Carlo/information gain estimation techniques(e.g., [6, 21, 51]).

Finally, consider that a single estimation of the causal discovery utility for an SCM with 20 nodes with $N = 500$ previously collected experimental samples takes approximately 2 minutes on an off-the-shelf laptop. Thus, for 10 BO iterations we can do the experimental design phase in 20 minutes (assuming we parallelize the utility optimization for each node). In a practical application scenario one might be very willing to invest hours or days for the design phase before conducting a costly experiment.

### F.4 Implementation and Computing Resources

Our Python implementation uses the PyTorch [44], GPyTorch [18], CDT [32], SKLearn [47], NetworkX [22] and BoTorch [4] packages, which greatly eased our implementation efforts. All of our experiments were run on CPUs. We parallelise the experiment design by running the optimisation process for each candidate intervention set on a separate core.

# G Evaluation Metrics

In this section, we provide details on the metrics used to evaluate our method in Section 3 and Appx. H. In our experiments, we use (nested) Monte Carlo estimators to approximate intractable expectations.

**Kullback-Leibler Divergence.** We evaluate the inferred posterior over queries given observed data, $p(Y \mid \boldsymbol{x}^{1:t})$, to the true query distribution $p(Y \mid \mathcal{M}^{\star})$ using the Kullback-Leibler Divergence (KLD), i.e.,

$$\mathrm{KL}(p(Y \mid \mathcal{M}^{\star}) \| p(Y \mid \boldsymbol{x}^{1:t})) = \mathbb{E}_{Y \mid \mathcal{M}^{\star}} \left[ \log p(Y \mid \mathcal{M}^{\star}) - \log p(Y \mid \boldsymbol{x}^{1:t}) \right] \tag{G.1}$$

$$= \mathbb{E}_{Y \mid \mathcal{M}^{\star}} \left[ \log p(Y \mid \mathcal{M}^{\star}) - \log \mathbb{E}_{\mathcal{M} \mid \boldsymbol{x}^{1:t}} [p(Y \mid \mathcal{M})] \right]. \tag{G.2}$$

**Query KLD.** For $Y = X_5^{\mathrm{do}(X_3 = \psi)}$ with $\psi \sim p(\psi)$ we have

$$\text{Query KLD} = \mathbb{E}_{\psi} \left[ \mathrm{KL}(p^{\mathrm{do}(X_3 = \psi)}(X_5 \mid \mathcal{M}^{\star}) \| p^{\mathrm{do}(X_3 = \psi)}(X_5 \mid \boldsymbol{x}^{1:t})) \right] \tag{G.3}$$

$$= \mathbb{E}_{\psi} \left[ \mathbb{E}_{X_5 \mid \mathrm{do}(X_3 = \psi), \mathcal{M}^{\star}} \left[ \log p^{\mathrm{do}(X_3 = \psi)}(X_5 \mid \mathcal{M}^{\star}) - \log p^{\mathrm{do}(X_3 = \psi)}(X_5 \mid \boldsymbol{x}^{1:t}) \right] \right]. \tag{G.4}$$

To approximate the outer two expectations, we keep a fixed set of samples for each ground truth SCM to enhance comparability between different ABCI runs. For $p^{\mathrm{do}(X_3 = \psi)}(X_5 \mid \boldsymbol{x}^{1:t})$, we use the estimator described in Section E.1.

**SCM KLD.** For $Y = q_{\mathrm{CML}}(\mathcal{M}) = \mathcal{M}$, we have

$$\text{SCM KLD} = \mathrm{KL}(p(\mathcal{M} \mid \mathcal{M}^{\star}) \| p(\mathcal{M} \mid \boldsymbol{x}^{1:t})) \tag{G.5}$$

$$= \mathbb{E}_{\mathcal{M} \mid \mathcal{M}^{\star}} \left[ \log p(\mathcal{M} \mid \mathcal{M}^{\star}) - \log p(\mathcal{M} \mid \boldsymbol{x}^{1:t}) \right] \tag{G.6}$$

$$= 0 - \log p(\mathcal{M}^{\star} \mid \boldsymbol{x}^{1:t}) \tag{G.7}$$

$$= -\log \mathbb{E}_{\mathcal{M} \mid \boldsymbol{x}^{1:t}} [p(\mathcal{M}^{\star} \mid \mathcal{M})] \tag{G.8}$$

$$= -\log \mathbb{E}_{G, \boldsymbol{f}, \boldsymbol{\sigma}^2 \mid \boldsymbol{x}^{1:t}} \left[ p(G^{\star}, \boldsymbol{f}^{\star}, \boldsymbol{\sigma}^{2,\star} \mid G, \boldsymbol{f}, \boldsymbol{\sigma}^2) \right] \tag{G.9}$$

$$= -\log \mathbb{E}_{G, \boldsymbol{f}, \boldsymbol{\sigma}^2 \mid \boldsymbol{x}^{1:t}} \left[ p(G^{\star} \mid G) \, p(\boldsymbol{f}^{\star} \mid \boldsymbol{f}) \, p(\boldsymbol{\sigma}^{2,\star} \mid \boldsymbol{\sigma}^2) \right]. \tag{G.10}$$

Now note that $p(G^{\star} \mid G) = 1$ if $G = G^{\star}$ and 0 otherwise, $p(\boldsymbol{f}^{\star} \mid \boldsymbol{f}) = \delta(\boldsymbol{f}^{\star} - \boldsymbol{f})$, and $p(\boldsymbol{\sigma}^{2,\star} \mid \boldsymbol{\sigma}^2) = \delta(\boldsymbol{\sigma}^{2,\star} - \boldsymbol{\sigma}^2)$. Hence, the SCM KLD vanishes iff the SCM posterior $p(\mathcal{M} \mid \boldsymbol{x}^{1:t})$ collapses onto the true SCM $\mathcal{M}^{\star}$, and is infinite otherwise.

**Average Interventional KLD.** Computing the KLD for $Y = q_{\mathrm{CML}}(\mathcal{M}) = \mathcal{M}$ is not useful for evaluation, since it vanishes when the SCM posterior $p(\mathcal{M} \mid \boldsymbol{x}^{1:t})$ collapses onto the true SCM $\mathcal{M}^{\star}$ and is infinite otherwise. For this reason, we report the *average interventional KLD* as a proxy metric, which we define as

$$\text{Avg. I-KLD} = \frac{1}{d} \sum_{i=1}^{d} \mathbb{E}_{\psi} \left[ \mathrm{KL}(p^{\mathrm{do}(X_i = \psi)}(X \mid \mathcal{M}^{\star}) \| p^{\mathrm{do}(X_i = \psi)}(X \mid \boldsymbol{x}^{1:t})) \right] \tag{G.11}$$

$$= \frac{1}{d} \sum_{i=1}^{d} \mathbb{E}_{\psi} \left[ \mathbb{E}_{X \mid \mathrm{do}(X_i = \psi), \mathcal{M}^{\star}} \left[ \log p^{\mathrm{do}(X_i = \psi)}(X \mid \mathcal{M}^{\star}) - \log p^{\mathrm{do}(X_i = \psi)}(X \mid \boldsymbol{x}^{1:t}) \right] \right] \tag{G.12}$$

$$= \frac{1}{d} \sum_{i=1}^{d} \mathbb{E}_{\psi} \Big[ \mathbb{E}_{X \mid \mathrm{do}(X_i = \psi), \mathcal{M}^{\star}} \Big[ \log p^{\mathrm{do}(X_i = \psi)}(X \mid \mathcal{M}^{\star}) \tag{G.13}$$

$$- \log \mathbb{E}_{\mathcal{M} \mid \boldsymbol{x}^{1:t}} \left[ p^{\mathrm{do}(X_i = \psi)}(X \mid \mathcal{M}) \right] \Big] \Big].$$

As with the Query KLD, we keep a fixed set of MC samples per ground truth SCM to approximate the two outer expectations to enhance comparability between different ABCI runs.

**Expected Structural Hamming Distance.** The Structural Hamming Distance (SHD)

$$\mathrm{SHD}(G, G^{\star}) = \left| \{(i,j) \in G : (i,j) \notin G^{\star}\} \right| + \left| \{(i,j) \in G^{\star} : (i,j) \notin G\} \right| \tag{G.14}$$

denotes the simple graph edit distance, i.e., it counts the number of edges $(i, j)$ that are present in the prediction graph $G$ and not present in the reference graph $G^\star$ and vice versa. We report the expected SHD w.r.t. our posterior over graphs as

$$\text{ESHD}(G, G^\star) = \mathbb{E}_{G \mid \boldsymbol{x}^{1:t}} \left[ \text{SHD}(G, G^\star) \right] \tag{G.15}$$

**AUPRC.** Following previous work [14, 16, 35, 65], we report the *area under the precision recall curve* (AUPRC) by casting graph learning as a binary edge prediction problem given our inferred posterior edge probabilities $p(G_{i,j} \mid \boldsymbol{x}^{1:t})$. Refer to e.g. Murphy [42] for further information on this quantity.

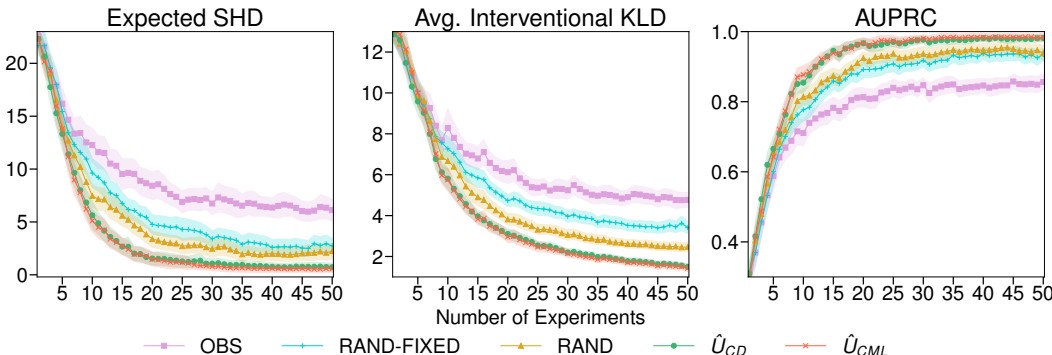

Figure 5: **Causal Discovery and SCM Learning on Scale-free Graphs with** 10 **Variables.** Comparison of the experimental design strategies with random and observational baselines on simulated ground truth models with 10 nodes. Lines and shaded areas show means and 95% confidence intervals (CIs) across 50 runs (10 randomly sampled ground-truth SCMs with 5 restarts per SCM). The $U_{CD}$ and $U_{CML}$ objectives perform on par with each other. Both clearly outperform the observational and random baselines on all metrics.

## H   Extended Experimental Results

**Causal Discovery and SCM Learning for SCMs with** $d = 10$ **Variables.**   We report results on ground truth SCMs with $d = 10$ variables and scale-free graphs in Fig. 5. We initialise all methods with 5 observational samples and perform experiments with a batch size of 3. All other parameters are chosen as described in Appx. E.

**Causal Discovery and SCM Learning for SCMs with** $d = 20$ **Variables.**   To demonstrate the scalability of our framework, we report results on ground truth SCMs with $d = 20$ variables and scale-free or Erdős-Renyi graphs in Fig. 6 and Fig. 7, respectively. We initialise all methods with 50 observational samples and perform experiments with a batch size of 5. All other parameters are chosen as described in Appx. E.

While ABCI shows clear benefits when scale-free causal graphs underlie the SCMs, we find that the advantage of ABCI diminishes on SCMs with unstructured Erdős-Renyi graphs, which appear to pose a harder graph identification problem. Moreover, we expect performance of our inference machinery, especially together with the informed action selection, to increase when investing more computational power to improve the quality of our estimates, e.g., by increasing the number of Monte Carlo samples used in our estimators and increasing the number of evaluations during the Bayesian optimisation phase.

Finally, in Fig. 8 we show that using a simple linear model (GP model with a linear kernel) is not able to reasonably capture the characteristics of the ground truth model (non-linear GP model) due to the model mismatch.

**Learning Interventional Distributions vs. Causal Discovery and SCM Learning.**   We report additional metrics for our causal reasoning experiment as described in § 3 in Figs. 9 and 10. The key result here is that $U_{CR}$ yields a significantly lower Query KLD while exhibiting a worse ESHD and Average I-KLD scores, which indicates that, indeed, the $U_{CR}$ learns only those parts of the model that are relevant to reducing the uncertainty in our target query. This is more data efficient than trying to learn the entire model first and then answering the causal query of interest.

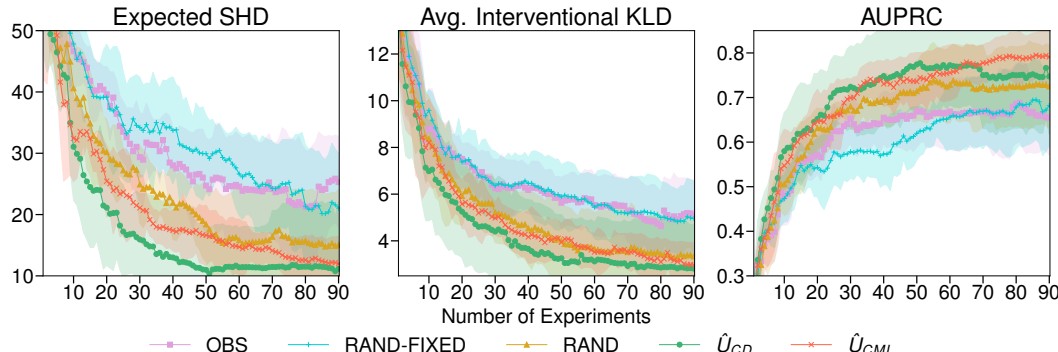

**Figure 6: Causal Discovery and SCM Learning on Scale-free Graphs with** 20 **Variables.** (Same figure as in Fig. 2 with additional confidence intervals for OBS and RAND FIXED.) Comparison of the experimental design strategy for causal discovery ($U_{\text{CD}}$) with random and observational baselines on simulated ground truth models with 20 nodes. Lines and shaded areas show means and 95% confidence intervals (CIs) across 15 runs (5 randomly sampled ground-truth SCMs with 3 restarts per SCM). The $U_{\text{CD}}$ objective significantly outperforms the observational and random baselines on all metrics.

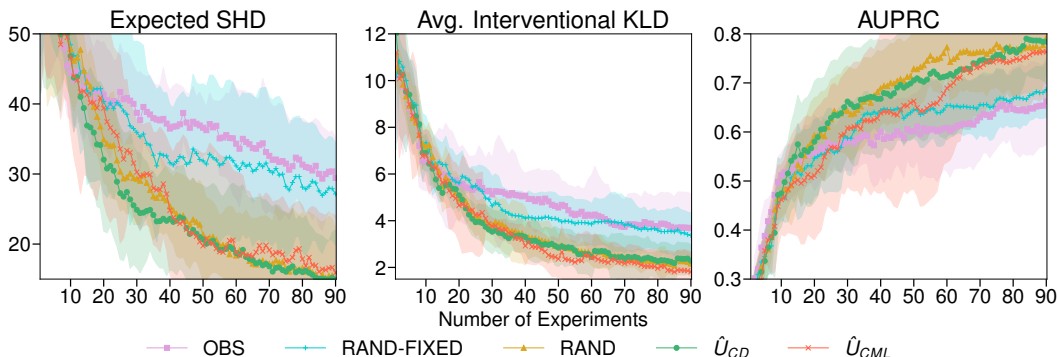

**Figure 7: Causal Discovery and SCM Learning on Erdős-Renyi Graphs with** 20 **Variables.** Comparison of experimental design strategies for causal discovery ($U_{\text{CD}}$) and causal model learning ($U_{\text{CML}}$) with random and observational baselines on simulated ground truth models with 20 nodes. Lines and shaded areas show means and 95% confidence intervals (CIs) across 15 runs (5 randomly sampled ground-truth SCMs with 3 restarts per SCM). The $U_{\text{CD}}$ and $U_{\text{CML}}$ strategies perform approx. equal to the strong random baseline (RAND) on all metrics, however, all three are significantly better than the weak random (RAND FIXED) and observational baselines. We expect that improving the quality of the $U_{\text{CD}}$ and $U_{\text{CML}}$ estimates (e.g., by scaling up computational resources invested in the MC estimates) yield similar benefits of the experimental design utilities as apparent in Fig. 6.

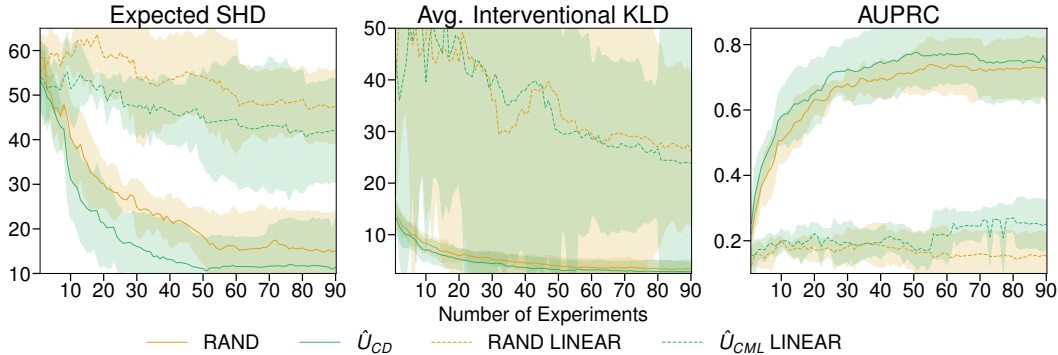

**Figure 8: Causal Discovery and SCM Learning on Scale-free Graphs with** 20 **Variables.** Comparison of non-linear GP model with a linear model (linear GP kernel) for $U_{\text{CD}}$ an RAND on simulated ground truth models with 20 nodes. Lines and shaded areas show means and 95% confidence intervals (CIs) across 15 runs (5 randomly sampled ground-truth SCMs with 3 restarts per SCM). Clearly, the model mismatch in the linear model prohibits the identification of the ground-truth graph.

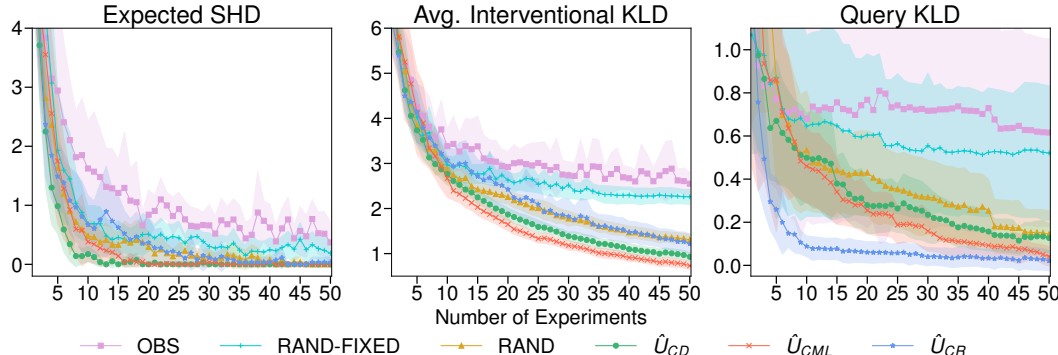

Figure 9: **Learning Interventional Distributions.** Comparison of the experimental design strategies with random and observational baselines. Lines and shaded areas show means and 95% confidence intervals (CIs) across 30 runs (10 randomly sampled ground-truth SCMs with 3 restarts per SCM). $U_{\mathrm{CD}}$, $U_{\mathrm{CML}}$ and $U_{\mathrm{CR}}$ perform best w.r.t. the ESHD, Avg. I-KLD and Query KLD metrics respectively, which is expected.

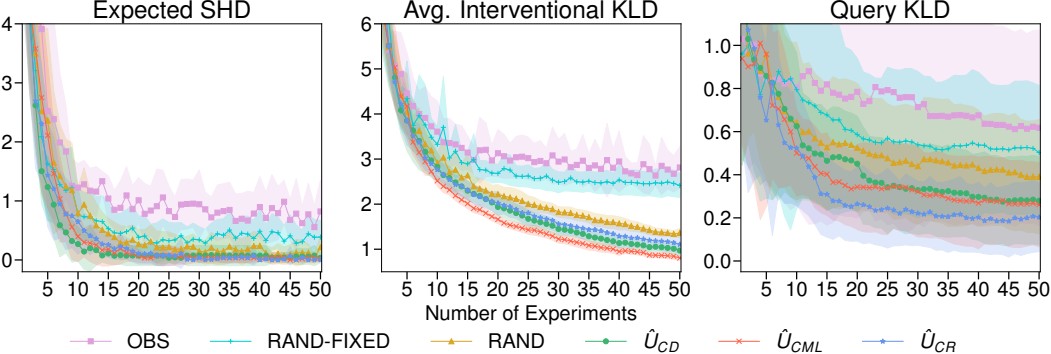

Figure 10: **Learning Interventional Distributions.** Comparison of the experimental design strategies with random and observational baselines. Lines and shaded areas show means and 95% confidence intervals (CIs) across 30 runs (10 randomly sampled ground-truth SCMs with 3 restarts per SCM). $U_{\mathrm{CD}}$, $U_{\mathrm{CML}}$ and $U_{\mathrm{CR}}$ perform best w.r.t. the ESHD, Avg. I-KLD and Query KLD metrics respectively, which is expected.

