# OpenReview forum: "Active Bayesian Causal Inference"
_NeurIPS.cc/2022/Workshop/nCSI — nCSI WS @ NeurIPS 2022 Poster_

### Official Review · Reviewer_vJsc · 2022-10-15
**Paper review**

**Rating:** 1
**Confidence:** 3

**Review:**

The paper is not in the adequate format for the workshop. The authors provide in the main part of the paper a general Bayesian formulation of problems in causal learning and inference. However, this formulation does not come with a general solution. More specifically, they show that a causal query can be solved by estimating its posterior given in Eq. 2.3, but it cannot be always estimated. One particular tractable implementation is proposed in Appendix A, but if this is the actual contribution of the paper, it should be the focus of the paper and explained in detail in the main part.

Moreover, the authors claim in line 102 that this view  “subsumes and generalises causal discovery and reasoning into a unified framework”. I strongly disagree with that. How can they claim that the proposed approach is more general if they only propose a solution for the class of causally sufficient nonlinear additive Gaussian noise models? An efficient approach that integrates causal discovery, causal inference, and experimental design is extremely desirable, but the paper does not propose a *general* solution for that.  The current “two-stage” approach of causal discovery followed by causal inference has indeed a lot of room for improvement, but it can be general in the sense that no strong assumptions about the model are necessary. For example, FCI accounts for the existence of unmeasured confounders and does not make any parametric or distributional assumptions about the model. Causal effect identification can then be determined from the learned Markov equivalence class (a PAG, in this case), thus accounting for all models that are compatible with the observed conditional independences in the data. To claim subsumption, the proposed framework should be able to solve this general class of models.

Minor: The description of the induced causal diagram in line 71 is incomplete as it does not explain the existence of dashed bidirected arrows between endogenous variables with a common exogenous parent.

---

### Official Review · Reviewer_2foL · 2022-10-15
**Principled Bayesian approach to integrating causal discovery and reasoning**

**Rating:** 2
**Confidence:** 2

**Review:**

**Overview**

The paper presents a novel Bayesian framework, ABCI, to integrate causal discovery and reasoning, which in prior works are addressed separately. The authors introduce the notion of a causal query that subsumes causal discovery, causal model learning and causal reasoning. The query posterior can then be computed to achieve the learning objective. In addition, the framework proposes actively collecting interventional data at each time step by first inferring a maximally informative intervention and then performing this intervention on the *true* causal model. The proposed approach is evaluated using simple data-generating processes and compared against baselines of collecting 1. observational data and 2. performing random interventions.

**Strengths**

Integrating causal discovery and reasoning is an important challenge and this paper presents a novel and principled approach to tackle it. Taking a Bayesian approach allows a natural and intuitive way to actively infer the best intervention to perform in each interation. The experimental results, although on a simple synthetic dataset, demonstrate the effectiveness of ABCI.

**Weaknesses**
A more thorough discussion on the restricted class of causal models for which the marginalization needed is computationally feasible would be useful. Also, the scalability of this approach needs to be addressed. Experiments on more complex data-generating processes might help in this regard.

Minor comment: The labels for the graphs in figure 3 seem to be missing.

Overall, this approach seems promising and potentially extensible to a wider class of causal models.

---

### Meta-Review · Program_Chairs · 2022-10-20

**Recommendation:** 2
**Confidence:** 3

**Metareview:**

One reviewer suggested accept, whereas the other reviewer suggested reject. Considering the former's review, what was mentioned as a way to improve was that an extended coverage of what makes the given model tractable would be desired. Something shared by the latter reviewer. However, the latter reviewer's main concern for rejection was the fact that the claim being made, regarding subsumption, is too strong. Still, both reviewers seem to recognize the value in the tractability portion of the analysis. Furthermore, as suggested by the former reviewer bridging formalisms (here, Bayesian formalism and causal inference) opens a new perspective that also becomes relevant for symbolic approaches. Therefore, accept is recommended.

---

### Decision · Program_Chairs · 2022-10-20

Accept (Poster)